# Adaptation and performance assessment of a Quantum / Interband Cascade Laser Spectrometer for simultaneous airborne in situ observation of $CH_4$, $C_2H_6$, $CO_2$, $CO$ and $N_2O$

Julian Kostinek[1], Anke Roiger[1], Kenneth J. Davis[3], Colm Sweeney[6], Joshua P. DiGangi[5], Yonghoon Choi[5,7], Bianca Baier[6,8], Frank Hase[4], Jochen Groß[4], Maximilian Eckl[1], Theresa Klausner[1], and André Butz[2]

[1]Deutsches Zentrum für Luft- und Raumfahrt, Institut für Physik der Atmosphäre, Oberpfaffenhofen, Germany
[2]Institute of Environmental Physics, University of Heidelberg, Heidelberg, Germany
[3]Department of Meteorology and Atmospheric Science, The Pennsylvania State University, University Park, PA 16802, USA
[4]Institute of Meteorology and Climate Research, Karlsruhe Institute of Technology, Karlsruhe, Germany
[5]NASA Langley Research Center, Hampton, VA 23681-2199, USA
[6]NOAA ESRL Global Monitoring Division, Boulder, CO 80305-3328, USA
[7]Science Systems and Applications, Inc., Hampton, VA 23681, USA
[8]Cooperative Institute for Research in Environmental Sciences, University of Colorado Boulder, Boulder, CO 80305, USA

**Correspondence:** Julian Kostinek (julian.kostinek@dlr.de)

**Abstract.** Tunable laser direct absorption spectroscopy is a widely used technique for in situ sensing of atmospheric composition. Aircraft deployment poses a challenging operating environment for instruments measuring climatologically-relevant gases in the Earth's atmosphere. Here, we demonstrate the successful adaption of a commercially available continuous wave quantum cascade (QCL) and interband cascade laser (ICL) based spectrometer for airborne in-situ trace gas measurements with a local to regional focus. The instrument measures methane, ethane, carbon dioxide, carbon monoxide, nitrous oxide and water vapor simultaneously, with high 1s-$1\sigma$ precision (740 ppt, 205 ppt, 460 ppb, 2.2 ppb, 137 ppt, 16 ppm, respectively) and high frequency (2 Hz). We estimate a total 1s-$1\sigma$ uncertainty of 1.85 ppb, 1.6 ppb, 1.0 ppm, 7.0 ppb and 0.8 ppb in $CH_4$, $C_2H_6$, $CO_2$, $CO$ and $N_2O$, respectively. The instrument enables simultaneous and continuous observations for all targeted species. Frequent calibration allows for a measurement duty cycle $\geq 90$ %. A custom retrieval software has been implemented and instrument performance is reported for a first field deployment during NASA's Atmospheric Carbon and Transport America (ACT-America) campaign in fall 2017 over the eastern and central US. This includes an inter-instrumental comparison with a calibrated cavity ring-down greenhouse gas analyzer (operated by NASA Langley Research Center, Hampton, USA) and periodic flask samples analyzed at the National Oceanic and Atmospheric Administration (NOAA). We demonstrate good agreement of the QCL/ICL based instrument to these concurrent observations within the combined measurement uncertainty after correcting for a constant bias. We find that precise knowledge of the $\delta^{13}C$ of the working standards and the sampled air is needed to enhance $CO_2$ compatibility when operating on the 2227.604 cm$^{-1}$ $^{13}C^{16}O_2$ absorption line.

# 1 Introduction

With steadily increasing greenhouse gas concentrations in the Earths atmosphere an improved understanding of the anthropogenic influence on climate is of major interest for the global civilization. Globally averaged carbon dioxide ($CO_2$) mole fractions have increased by 40 % since 1750. Methane ($CH_4$) mole fractions have more than doubled since the pre-industrial era and over 60 % of this increase is estimated to be of anthropogenic nature (Ciais et al., 2013). Nitrous oxide ($N_2O$) is a strong greenhouse gas and is expected to have the most important ozone-depleting anthropogenic impact throughout the 21st century (Ravishankara et al., 2009). Ethane ($C_2H_6$) is a powerful tracer commonly used to discriminate between different types of methane sources (Smith et al. (2015); Barkley et al. (2017); Peischl et al. (2015)) and carbon monoxide (CO) is a marker for incomplete combustion processes and relates to the formation of tropospheric ozone (Klemm et al., 1996).

Aircraft provide a flexible platform for satisfying the fundamental need for accurate, temporally and spatially dense observations of these climatologically-relevant gases from local to regional scales. On-board meteorological data acquisition systems allow for concurrent observations of important atmospheric state variables like the local wind field, which is particularly useful to estimate emissions. Spectroscopic instruments making use of molecular ro-vibrational absorption allow for high temporal coverage through fast instrument response times (Chen et al., 2010). Some have already been used for airborne research, e.g. established IR spectrometers (O'Shea et al. (2013); Santoni et al. (2014); O. L. Cambaliza et al. (2015); Filges et al. (2015)). Significant effort led to instruments operating in the mid infrared (IR) region, e.g. liquid nitrogen cooled lead-salt diode laser based spectrometers (Fried and Richter, 2007). With the commercial availability of continuous-wave lasers emitting in the mid IR region near ambient temperature (Capasso (2010); Vurgaftman et al. (2015); Kim et al. (2015), Beck et al. (2002)) several new instrument designs have emerged (McManus et al. (2015); Zellweger et al. (2016)). QCL/ICL based systems exploit several orders of magnitude stronger molecular absorption features in the mid infrared compared to near infrared based instruments. Richter et al. (2015) reported on a custom-built difference frequency generation (DFG) absorption spectrometer for simultaneous in-situ detection of formaldehyde ($CH_2O$) and $C_2H_6$ providing high detection sensitivities of 40 ppt and 15 ppt, respectively. The custom-built airborne QCL spectrometer described by Catoire et al. (2017) allows for simultaneous observation of CO, $CH_4$ and nitrogen dioxide ($NO_2$) with in-flight precisions of 0.3 ppb, 5 ppb and 0.3 ppb for a sampling time of 1.6 s. McManus et al. (2011) reported on the development of a high-sensitivity trace gas instrument based on quantum cascade lasers and astigmatic Herriott cells with up to 240 m path length. Unlike many established instruments measuring different species sequentially (one species after the other), the described spectrometer allows for concurrent sensing of the selected species and faster response times. These instruments have already been operated on different research aircraft. Santoni et al. (2014) describe the successful deployment and evaluation of a similar airborne spectrometer (Harvard QCLS) for more than 500 flight hours. However, Pitt et al. (2016) reported a severe cabin pressure dependency of their $N_2O$ and $CH_4$ measurements using a commercial instrument (Aerodyne QCLS). By implementing a pressure-differentiated calibration method they were able to correct the corresponding data set, but had to omit roughly half of the measured data. Recently, Gvakharia et al. (2018) reported on a similar cabin pressure dependency for their $N_2O$, $CO_2$ and CO measurements (based on an Aerodyne QCLS). They suggested a fast calibration procedure to overcome these dependencies while maintaining a $\geq$ 90 % duty cycle.

Here, we describe the setup and performance of our flight-proven (over 100 flight hours) airborne QCL/ICL system developed for simultaneous airborne measurements of $CH_4$, $C_2H_6$, $CO_2$, CO, $N_2O$ and $H_2O$. The instrument is shown to provide multi-species airborne observations for assessing greenhouse gas fluxes with a local (e.g. single facilities) to regional focus (e.g. urban agglomerations). Simultaneous observations of $CH_4$ and $C_2H_6$ facilitate to pinpoint sources of $CH_4$ enhancements (Smith et al.,

2015). At the same time, the instrument provides measurements of $N_2O$, the third most important greenhouse gas. This makes the instrument an ideal tool for airborne quantification and source attribution of greenhouse gas emissions using e.g. the aircraft based mass balance approach. Section 2 summarizes the refinements over the commercial system for use on aircraft. We show that frequent two-point calibration can mitigate cabin pressure dependencies. Section 3 describes our custom-built retrieval software developed for tuning the retrieval process. Sections 4 and 5 report on instrument performance in the laboratory and in

the field during NASA's ACT-America fall 2017 campaign, including an inter-instrumental comparison with a calibrated cavity ring-down instrument and periodically taken flask samples. Section 6 summarizes our findings and concludes the study.

## 2    The airborne DLR QCL/ICL spectrometer

The spectrometer system used here builds upon the *Dual Laser Trace Gas Monitor*, a commercial tunable IR laser direct absorption spectrometer (TILDAS) available from *Aerodyne Research Inc., Billerica, USA*, acquired by *Deutsches Zentrum*

*für Luft- und Raumfahrt* (DLR) in late 2016. The basic instrument has already been extensively described in McManus et al. (2011). We will therefore only briefly introduce the basic instrument setup followed by a description of the refinements required to operate the instrument on a research aircraft.

### 2.1    Basic instrument setup

The spectrometer is split into an electronics and an optics compartment. The electronics compartment includes an embedded

computing system, thermoelectric cooling (TEC) controllers, power supplies, etc. The optics compartment includes the lasers, the sample cell, the pressure controller, etc. Figure 1 shows a top-view photograph of the optics compartment. A combination of a continuous wave (CW) QCL and ICL measures mole fractions of $CH_4$, $C_2H_6$, $CO_2$, CO, $N_2O$ and $H_2O$ simultaneously by direct absorption spectroscopy. The sample cell is an astigmatic Herriott cell with approximate physical dimensions of 15 cm x 15 cm x 50 cm (WxHxL) made from aluminum. It provides an effective absorption path length of 204 m with a net volume

of 2.1 L. Two laser light sources are tuned to a specific center wavelength by adjusting the operating temperature using Peltier elements contained in the lasers housing. Excess heat is removed through a liquid cooling/heating circuit (*Solid State Cooling Systems, New York, USA*). Laser #1 is an Interband cascade laser manufactured by *Nanoplus GmbH, Gerbrunn, Germany* with a peak output power of 9.5 mW operated at 4.7 °C and modulated between 2988.520 cm$^{-1}$ and 2990.625 cm$^{-1}$ using a linear current ramp of up to 40 mA. Laser #2 is a quantum cascade laser manufactured by *Alpes Laser, St-Blaise, Switzerland* with

a peak output power of 40 mW operated at 1.5 °C modulated between 2227.550 cm$^{-1}$ and 2228.000 cm$^{-1}$ using a linear current ramp of up to 300 mA. The lasers are modulated sequentially at a fixed frequency of 1.5 kHz. Laser #1 scans over absorption lines of $CH_4$, $C_2H_6$ and $H_2O$, Laser #2 sweeps over $N_2O$, $CO_2$ and CO lines. Each laser is sampled at 450 spectral

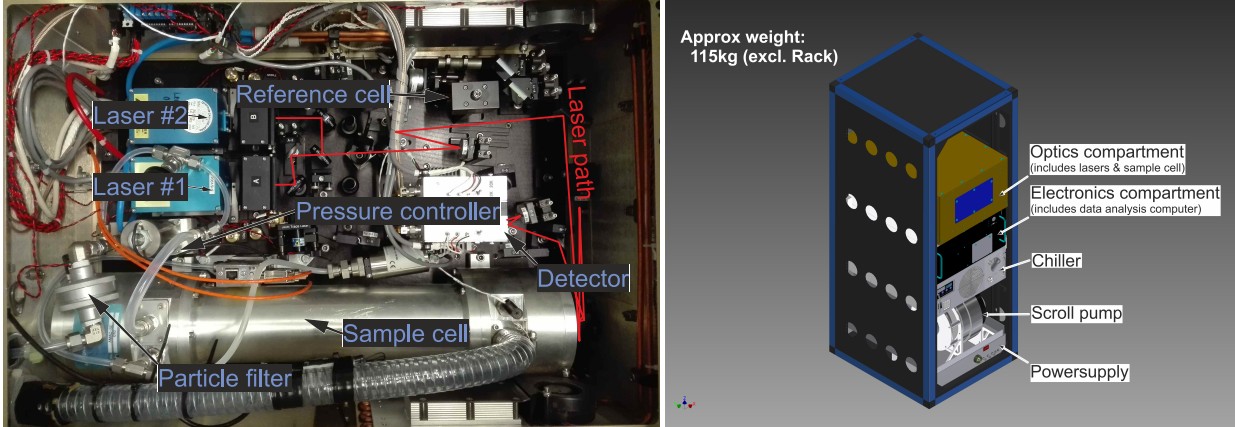

**Figure 1.** Top-down photograph on the optics compartment (left panel). The sample cell made from aluminum along with the pressure controller and pressure transducers can be identified in the lower half. The QCL/ICL lasers are mounted inside the blue housings to the left of the collimating Schwarzschild telescopes in the two black housings. The two detectors are mounted below the silver aluminum cases, housing the pre-amplifiers, on the right. The first detector is used for detecting both lasers after passing through the sample cell. The second detector is used for spectral referencing through an auxiliary optical path. The right panel illustrates the rack mounted instrument. The figure includes solid models from Aerodyne Inc. and Solid State Cooling Systems.

points. Acquired spectra are co-added to yield a single output spectrum twice per second. Before reaching the sample cell, the laser beam travels approximately 1.6 m inside the instrument under ambient conditions. This will be referred to as the open-path of the instrument, which is heavily influenced by variations in cabin pressure, temperature and humidity during airborne operation. After passing through the sample cell, the combined output from both lasers hits a single TEC-cooled detector. A second, identical detector collects radiation from two auxiliary paths. The first auxiliary path contains a small, sealed reference cell filled with $CH_4$ and $N_2O$. This allows for spectral referencing during system startup. The second path introduces an etalon into the beam, allowing for experimental determination of the laser tuning rate, which relates laser supply current and emitted wavelength.

## 2.2 Refinements for airborne operation

The key challenges for a successful deployment on research aircraft are limited space and power, the occurrence of linear and angular accelerations and large pressure, temperature and humidity fluctuations in both cabin and sampled air. Airborne instrumentation further requires a fast system response time, owing to the rapid movement of aircraft in the atmosphere. The response time is controlled by the time it takes to completely exchange the air in the sample cell which is driven by the highest achievable volumetric flow rate given a specific pump and sample cell volume.

Here, a scroll pump has been chosen to enable a constant sample flow through the sample cell. The lubricant-free scroll pump runs very smoothly, reducing vibrations of the measurement system, yet providing good pumping performance with a nominal value of 500 liters per minute at standard conditions. This translates to a net flow rate of 25 SLPM (given IUPAC standard

conditions of $T = 273.15$ K and $p = 1000$ hPa) when operating at a cell pressure of 50 hPa. Earlier experience showed that large electrical inrush currents have jeopardized nominal system startup (priv. comm. Stefan Müller, MPI Mainz). Sudden power failure, due to over-current triggering aircraft circuit breakers, may lead to failures in the data analysis equipment. The original motor has therefore been exchanged with a synchronous three-phase motor (*Baumueller Nuernberg GmbH, Velbert, Germany*). This DC motor provides a rated power of 627 W at 28 VDC. By using a digital motor controller the maximum startup current can be limited amongst various other tuning options. From previous studies the motor is known to emit a considerable amount of heat; a forced airflow provided by a standard axial fan ensures motor temperatures stay in the rated range.

Aircraft deployment requires the entire system to operate with a maximum of 50 A at 28 VDC. Power consumption of the instrument is mainly dominated by the pump and the thermoelectric cooling making up more than 3/4 of the total power requirement. Both components have been electrically converted without the need for power inverters from 230 VAC to 28 VDC to increase overall efficiency. The spectrometer and its internal computer are driven by a power inverter.

Large parts of the wiring harness have been exchanged from standard PVC cables to aviation-grade fire-resistant wiring. Mandatory electromagnetic compatibility/interference (EMC/EMI) tests have been carried out to comply with Federal Aviation Administration (FAA) regulations. The rack-mounted instrument sums up to a total mass of approximately 115 kg and has been tested to withstand linear accelerations of up to 9 g on the aircraft forward axis, 8 g on the downward axis, 6 g on the upward and 2.25 g sidewards. Due to aircraft certification issues, pure water is used as process fluid for the liquid cooling/heating circuit instead of the intended propylene glycol / water mixture.

A 3/8" inner diameter hose made out of polytetrafluoroethylene (PTFE) has been chosen for the sample air intake as a compromise between pressure drop across the inlet and to minimize lag time between the inlet and the sample cell. Inside the instrument and upstream of the sample cell, an aerosol filter holds back particles bigger than 2 $\mu$m. The inlet is rear facing, preventing large particle entrainment and protecting the instrument from liquid water and ice. Owing to the small diameter, the intake flow is inside the turbulent regime at all times ($Re \sim 4000$).

Finally, the sample cell pressure is regulated by means of a fully-configurable pressure controller (*Bronkhorst High-Tech B.V., Ruurlo, Netherlands*). A chip-scale temperature-compensated pressure transducer (*Measurement Specialties (Europe), Ltd.*) and a humidity sensor (*Sensirion AG, Staefa ZH, Switzerland*) have been built into the optics compartment, to allow for monitoring the open path state variables (see section 2.1).

## 2.3   In-flight calibration strategy

A custom-built calibration system has been implemented as illustrated in Figure 2. Using mass flow controllers (MFCs, *Bronkhorst High-Tech B.V., Ruurlo, Netherlands*), two gases can be mixed at arbitrary ratios. The calibration gas mixture has been chosen to resemble "target" gas mole fractions close to atmospheric ambient values. The cylinders have been cross-calibrated against NOAA standards using a cavity ring-down spectrometer (CRDS) and are thus traceable to World Meteorological Organization (WMO) standards for $CH_4$ (Cert.-Nr. CB11361, WMO X2004A for $CH_4$ (Dlugokencky et al., 2005), WMO X2007 for $CO_2$ (Zhao and Tans, 2006)). $C_2H_6$, CO and $N_2O$ are compared to NOAA flask samples taken during the

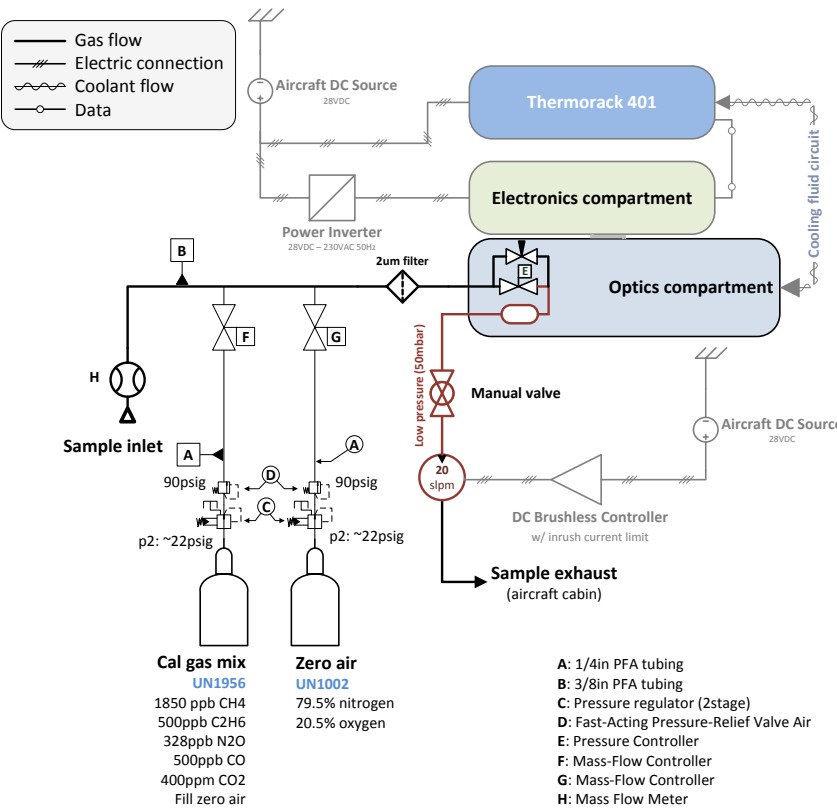

**Figure 2.** Schematic showing the main components with emphasis on the calibration system. A mass flow meter allows for measuring the sample flow rate. Two reference gases can be mixed at any arbitrary ratio by means of two calibrated mass flow controllers. A $2\mu m$ particle filter upstream of the sample cell avoids cell contamination.

ACT-America field campaigns, which are also traceable to WMO standards. We use ultra-pure synthetic air as "zero" gas instead of pure nitrogen ($N_2$) to be in accordance with aircraft safety regulations and because the mole fraction of synthetic air (79.5 % $N_2$ and 20.5 % $O_2$) is chemically closer to sampled atmospheric air. Our calibration setup allows the net flow rate from the calibration cylinders to be slightly higher than the sample flow rate, minimizing pressure variations in the sample cell
5  during switchover from normal to calibration sampling. To avoid contamination with cabin air, leak tests have been carried out on a regular basis during the ACT-America field campaign. Histograms of typical calibration measurements are provided in the Supplement Section 4.

Owing to the high sensitivity of the retrieved mole fractions to changes in ambient conditions during flights (Gvakharia et al., 2018), calibration cycles are carried out automatically every 5 to 10 minutes. Each cycle consists of a pre-programmed sequence
10  of flushing the sample cell with zero gas for 10 seconds followed by another 10 seconds of calibration gas. These time intervals have been found to be a good compromise between calibration gas cylinder endurance and measurement duty cycle. The online

mixing feature is not used for in-flight calibration. Hence, no dilution of the calibration standard with zero air is introduced during flights and the uncertainty in the flow rate measurements can be omitted. Online mixing (relevant for linearity checks) adds the uncertainty of the controlled mass flow (0.5 % relative error) on top of the gas cylinder uncertainties. Measured mole fractions of all detected species settle to an approximately constant value within the first two seconds after switchover from

5    calibration gas to sample air and vice versa. The only exception is water vapor, which is observed to settle after approximately 30 seconds because of its stickiness and because the inlet tubing is made out of PTFE. The observed decay in $H_2O$ is different from the decay in other species in that a slow, almost linear decay follows the initial exponential decay, due to remaining water vapor in the inlet tubing and the sample cell.

## 3    Data Retrieval & Post-processing

10    The standard approach to retrieve dry-air mole fractions from the Aerodyne QCLS instruments is by making use of the software supplied by the manufacturer (TDLWintel). Here we utilize a custom retrieval software (JFIT) developed to double check the output of the TDLWintel software and to enhance the ability of tweaking the retrieval process. Our main goal in developing a stand-alone algorithm here, was to learn about possible error sources, mitigation possibilities of instrument dependencies and to be able to extend the instruments capabilities in the future.

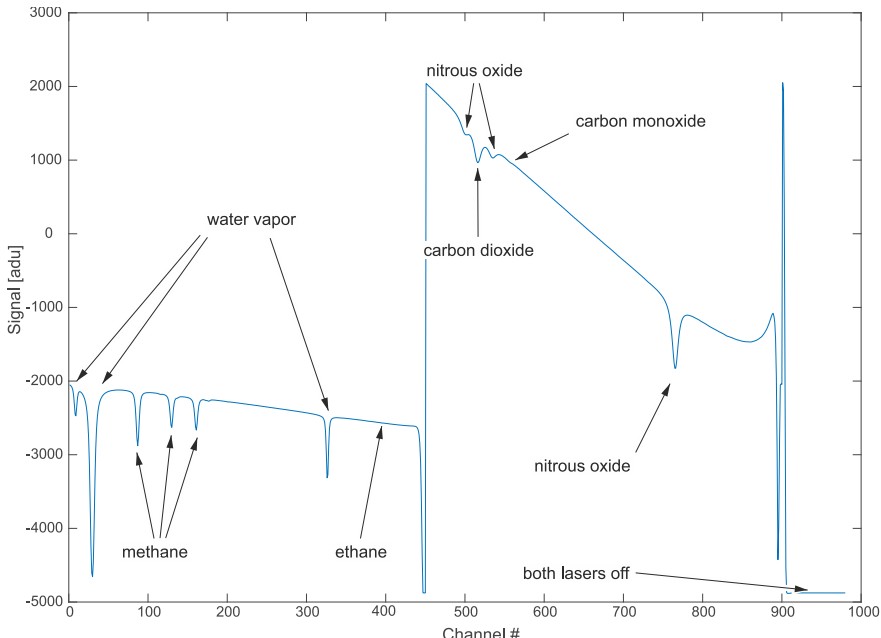

**Figure 3.** A typical raw spectrum as recorded in binary format by the instrument. Arrows have been added to ease identification of the observed chemical species. Channel numbers on the abscissa can be converted to spectral units using the laser tuning rate. The intensity offset can be corrected by shifting the entire spectrum to yield zero intensity when lasers are turned off.

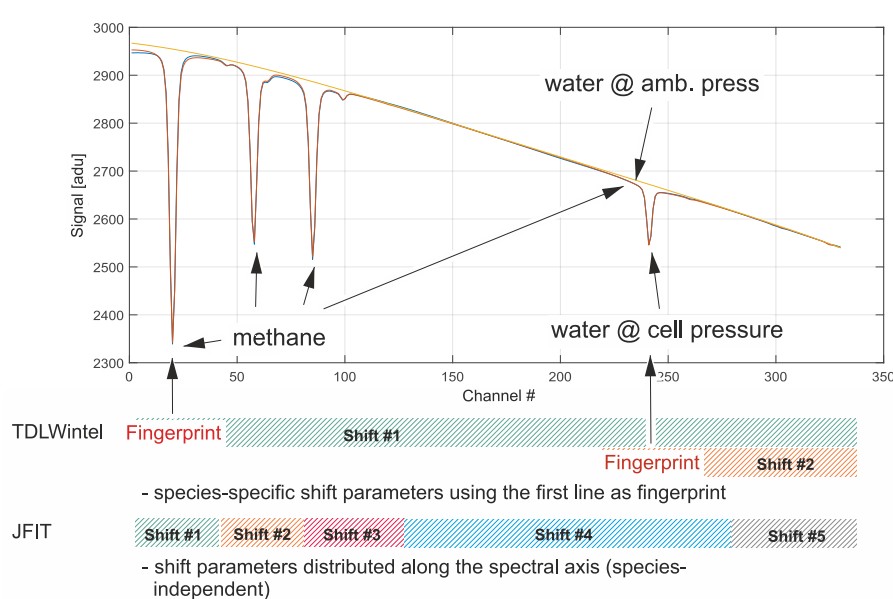

**Figure 4.** Schematic depicting the handling of spectral shift parameters and baseline modeling. The spectral baseline is fitted as a polynomial together with absorption features over the entire fit window. Shift parameters have been implemented in a species-independent way. Open-path water is also included in the model.

The code is written in plain C++. It digests the sample cell pressure and temperature measurements to generate a synthetic spectrum based on line-by-line parameters from the HITRAN2012/HITRAN2016 (Rothman et al. (2013); Gordon et al. (2017)) database using a conventional Voigt profile approach. Ethane line-by-line data have been taken from high-resolution FTIR spectra due to deficiencies in the HITRAN data for this particular species/wavenumber combination (Harrison et al., 2010). The computation of the Voigt profile has been adopted from Abrarov and Quine (2015). Our retrieval code differs from the TDLWintel approach in the determination of the spectral baseline, the handling of shift parameters and open path water absorption.

A typical raw spectral output, as saved by the instrument in binary format is illustrated in Figure 3. The two consecutive laser scans are clearly visible. On the left side, Laser #1 sweeps between $2988.520$ cm$^{-1}$ and $2990.625$ cm$^{-1}$ and hence, over absorption features of $CH_4$, $C_2H_6$ and $H_2O$. The right side corresponds to the wavelength range of Laser #2 ($2227.550$ cm$^{-1}$ to $2228.000$ cm$^{-1}$) and includes absorption features of $N_2O$, $CO$ and $CO_2$. After the lasers have scanned their full range, both lasers are completely turned off to allow for the determination of the detector zero-intensity offset. The abscissa corresponds to the individual sampling points, which can be converted to spectral units using the known laser tuning rate. The flat sections of the spectrum with no molecular absorption, are considered to represent the spectral baseline. The shape of this baseline is mainly controlled by laser characteristics, the detector response function and optical properties of the installed mirrors and windows inside the instrument.

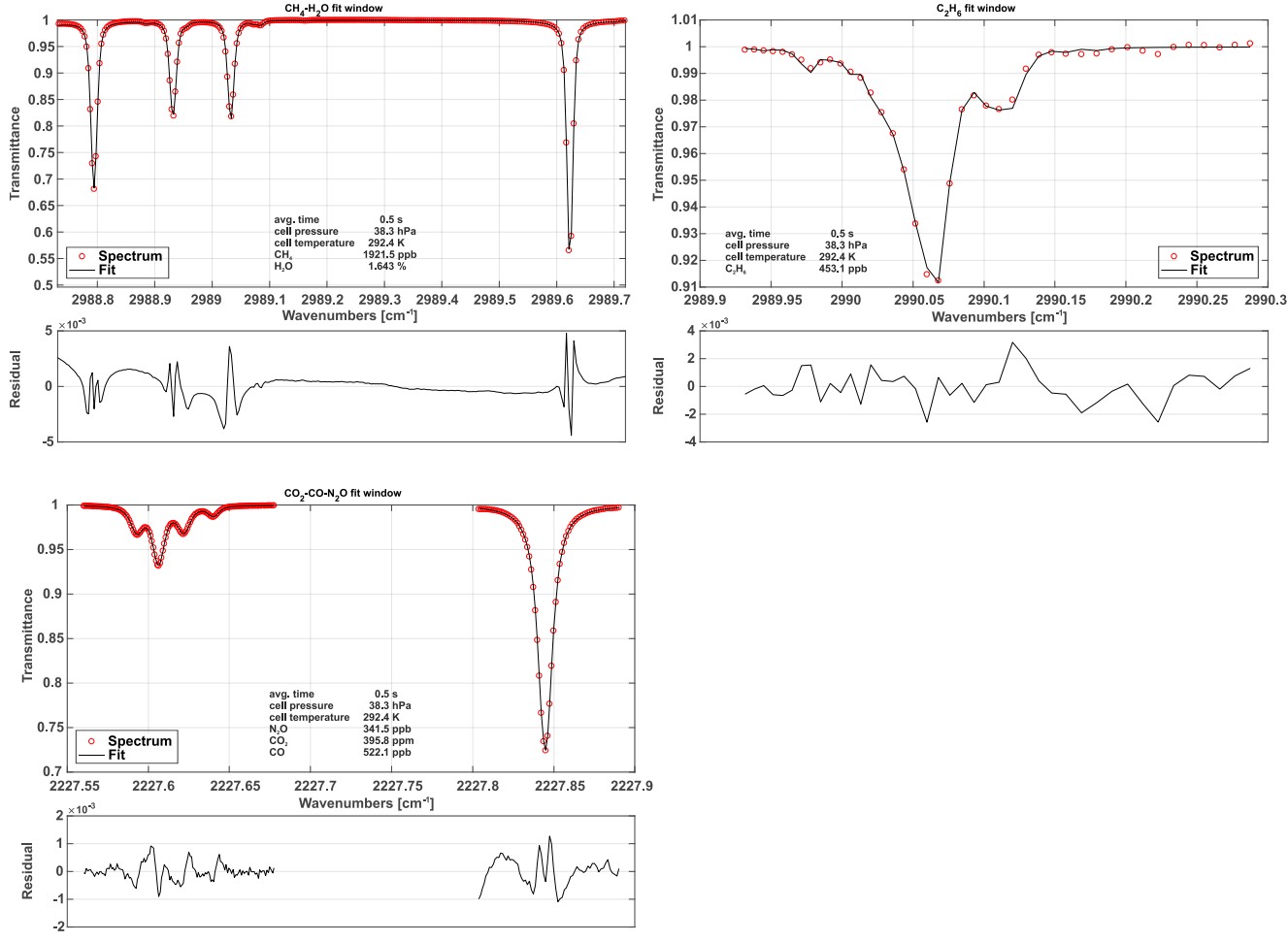

**Figure 5.** Typical, normalized spectra for each fit window including fits and associated residuals. The first fit window (top left) includes $CH_4$ and $H_2O$ absorption features. The top right fit window depicts $C_2H_6$ absorption. The lower left spectrum shows CO, $CO_2$ and $N_2O$ absorption.

The spectrum is broken down into 3 fit windows for the retrieval process (see Figure 5). These were chosen based on the best overall performance found in retrieval tests and named after the chemical species included. A synthetic spectrum, including a polynomial representing the spectral baseline, is generated and fitted using an unbounded Levenberg-Marquardt least-squares algorithm (Marquardt, 1963). The degree of the background-fitting polynomial has been adjusted empirically for each different fit window. Species independent shift parameters have been included allowing individual absorption features to freely move on the spectral axis. Special care has been taken to group weak and strong absorption features together in a single shift parameter, to provide sufficient certainty on their spectral positions. In other words, not every absorption line has its own shift parameter, but they are grouped as schematically shown in Figure 4. As a result, only 5 shift parameters are included although the synthetic

spectrum in Figure 4 is composed of more than 20 individual lines. When the absorptivity does not yield enough certainty to ensure proper determination of the shift parameters for a single spectrum, the shift variables are held constant at their means over the last ten values. If another species in the relevant fit window allows for a proper determination of the spectral position, remaining shift parameters are coupled to those with enough certainty. This allows to properly model absorption line center

frequency changes and provides a means for observing spectral stability. Typical shift parameters for ground-based operation are given in Figure 8 for the $CH_4$-$H_2O$ and $CO_2$-$CO$-$N_2O$ fit windows. Pressure, humidity and temperature data obtained from within the optics compartment are used to model $H_2O$ absorption at cabin pressure in the open-path region.

The $CH_4$-$H_2O$ fit window covers almost the entire set of spectral features covered with Laser #1 except for the $C_2H_6$ absorption features. The spectral baseline is modeled as a third-order polynomial over the full range of the fit window. A typical spectrum

including fit is depicted in Figure 5 along with typical spectra for the other two fit windows.

The $C_2H_6$ fit window includes absorption features of $CH_4$ and $C_2H_6$. The main challenge of retrieving precise $C_2H_6$ mole fractions arises from its very low background concentration in the atmosphere (approximately $1.05$ ppb in the northern hemisphere (Simpson et al., 2012)). A single adjacent $CH_4$ line, located at $2989.981$ cm$^{-1}$ has been included in order to obtain $C_2H_6$ data even under these challenging conditions. In this case, the weak $CH_4$ absorption is not modeled as a free parameter

and is hence not used for retrieving the $CH_4$ mole fraction, but for localizing the spectral position / shift parameter of the $C_2H_6$ absorption feature in the absence of a clear $C_2H_6$ signal. The $CH_4$ mole fractions are fixed to the values determined from the previous fit window. Using this approach, we found a clear improvement in the $C_2H_6$ data quality including a higher precision and the absence of discontinuities. The associated spectral baseline is modeled as a second-order polynomial.

The $CO_2$-$CO$-$N_2O$ fit window covers the entire second laser and is the most complex spectral scene. It includes several over-

lapping absorption features making the retrieval of mole fractions of the targeted species challenging. As illustrated in Figure 5, a single $CO_2$ absorption line is surrounded by two $N_2O$ lines. The $CO$ line is directly adjacent to one of the $N_2O$ lines. This results in comparatively large signal noise and increased uncertainty in the retrieved mole fractions due to crosstalk between the $N_2O$, $CO$ and $CO_2$ absorption lines. However, the spectral range includes another $N_2O$ line at $2227.843$ cm$^{-1}$, which is slightly stronger than the other two (see Figure 5). Our approach is to fix the mole fractions of the first two $N_2O$ lines to the

stronger third one, in order to reduce the uncertainty in retrieved $N_2O$ and hence the noise on the $CO_2$ and $CO$ retrieval. The spectral baseline has been split into two parts, the first covering the first two $N_2O$, $CO_2$ and $CO$ lines, and the second covering the individual $N_2O$ line only. Both are modeled as second-order polynomials.

## 3.1 Water vapor correction

In the current instrument setup, water vapor is not removed from sampled air before entering the sample cell. Therefore, the

influence of water vapor on the retrieved mole fractions has to be corrected in order to report dry-air mole fractions. Here, we correct for both, dilution and water broadening effects. The first describes the fact that concentrations appear smaller when analyzing moist air, although the dry air mole fraction might be constant. This effect can be remedied for if the absolute water

concentration is known for each individual sample using Eq. (1)

$$c_d = \frac{c_x}{(1 - c_{H_2O})} \tag{1}$$

where $c_d$ is the dry-air mole fraction, $c_x$ is the raw concentration of a particular species of interest diluted in moist air and $c_{H_2O}$ is the water vapor concentration (Harazono et al., 2015). Spectroscopic water broadening effects are approximately an order of magnitude smaller than dilution effects, yet they do have to be corrected for to obtain precise measurements. HITRAN's air broadening parameters are listed for a particular chemical composition of air excluding water vapor. $H_2O$, however, can be a more potent broadening agent than nitrogen or oxygen (Kooijmans et al., 2016). These coefficients have been determined using the setup depicted in Figure 6 and are summarized in Table 1. Therefore, the pressure broadening has to be modified to include this effect. Under dry air conditions it is common to split the pressure broadening into two parts: self-broadening and air-broadening. The self-broadening coefficient allows computation of the broadening induced by mutual collisions of a particular species of interest. The air-broadening coefficient can be used to approximate the broadening induced through collisions of a particular species with all the other species in a given air standard excluding the species itself. From the HITRAN definitions,

**Table 1.** Empirically determined water vapor foreign broadening coefficients

| Chemical species | $CH_4$ | $C_2H_6$ | $CO_2$ | $CO$ | $N_2O$ |
|---|---|---|---|---|---|
| Broadening coefficient ($\gamma_{air}$) | 1.05 | 1.18 | 2.2 | 2.1 | 2.2 |

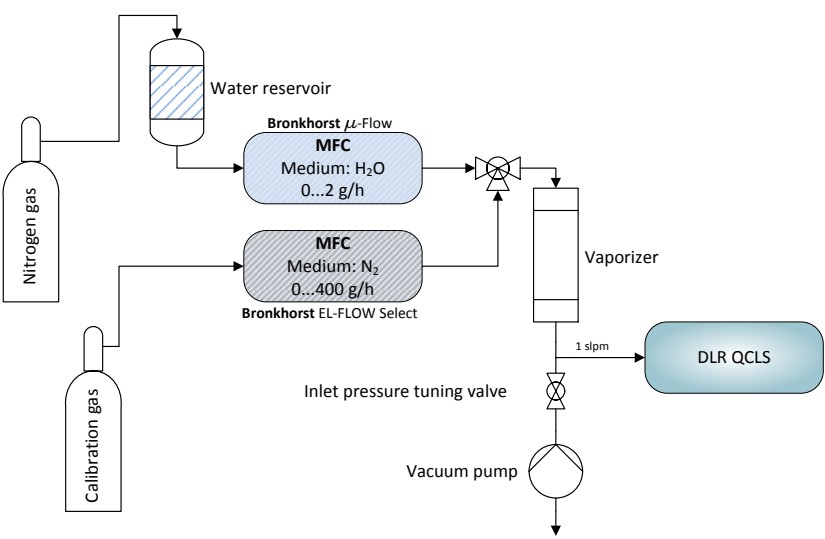

**Figure 6.** Schematic depicting the water correction lab setup. A reference gas can be humidified to typical atmospheric values between 0 % and 2 % absolute water using mass flow controllers and an electronically controlled vaporizer. A downstream pump allows for simulation of different flight levels.

the pressure-broadened half width at half maximum for a gas at pressure $p$ and temperature $T$ is given by

$$\gamma\left(p, T\right) = \left(\frac{T_{ref}}{T}\right)^{n_{air}} \left(\gamma_{air}\left(p - p_{self}\right) + \gamma_{self} p_{self}\right) \tag{2}$$

where $T_{ref}$ is a fixed reference temperature ($T_{ref} = 296$ K), $p_{self}$ is the partial pressure of a particular species of interest and $n_{air}$ is the coefficient of the temperature dependence of the air-broadened half width. This model has been extended to include
collisions with $H_2O$ molecules yielding

$$\gamma\left(p, T\right) = \left(\frac{T_{ref}}{T}\right)^{n_{air}} \left(\gamma_{air}\left(p - p_{self} - p_{H_2O}\right) + \gamma_{self} p_{self} + \gamma_{H_2O} p_{H_2O}\right) \tag{3}$$

with the partial pressure of water vapor $p_{H_2O}$ and the water broadening coefficient $\gamma_{H_2O}$. The former can be computed from the measured water vapor concentration. The latter can be empirically determined. Not including the self and water foreign broadening leads to relative errors in the range of 0-2 % for the described setup, depending on the species of interest. While
small for $C_2H_6$ and $CH_4$ with $< 0.03$ %, the influence on retrieved CO is rather large with $\sim 2$ %. In order to obtain $\gamma_{H_2O}$ two MFCs are used to modify mole fractions of water vapor in a clean and dry calibration gas. This does not involve measuring water vapor at absolute levels, instead it is only necessary to span the range of atmospheric $H_2O$. An additional downstream pump allows, in combination with a manually-controlled needle-valve, tuning the absolute pressure at the instrument inlet to simulate altitude changes. For these tests, the QCLS instrument has been operated at low flow rates of approximately 1 SLPM
due to limitations on the two mass flow controllers. The water broadening coefficient $\gamma_{H_2O}$ has been adjusted iteratively until reported dry-air mixing-ratios of the species of interest remained constant for the set of water vapor mole fractions.

## 4   Ground-based performance

Extensive ground-based instrument checks have been conducted, including tests in a pressure chamber at the Karlsruhe Institute of Technology (KIT) and laboratory tests at DLR Oberpfaffenhofen, Germany. These tests confirmed the presence of
an ambient pressure dependence found in earlier studies (i.e. Pitt et al. (2016)). Here, we show in-field, ground-based instrument checks conducted in Hangar N-159 at NASA Wallops Flight Facility, Wallops Island, USA, to ensure proper instrument operation and determine instrument precision. Electric current drawn from the aircraft remained under 50 A at all times and settled at approximately 40 A. The flow rate stabilized at 23 SLPM for a sample cell pressure regulated at $50.0 \pm 0.5$ hPa (0.2 hPa precision @ 5 Hz frequency). Typical precision (standard deviation for 1 s averaging) for ground-based operation is
summarized in Table 2.

**Table 2.** Typical $1s - 1\sigma$ precision during ground-based instrument checks.

| Chemical species | $H_2O$ | $CH_4$ | $C_2H_6$ | $CO_2$ | CO | $N_2O$ |
|---|---|---|---|---|---|---|
| Precision $1s - 1\sigma$ | 2.1 ppm | 142 ppt | 87 ppt | 169 ppb | 1.3 ppb | 45 ppt |

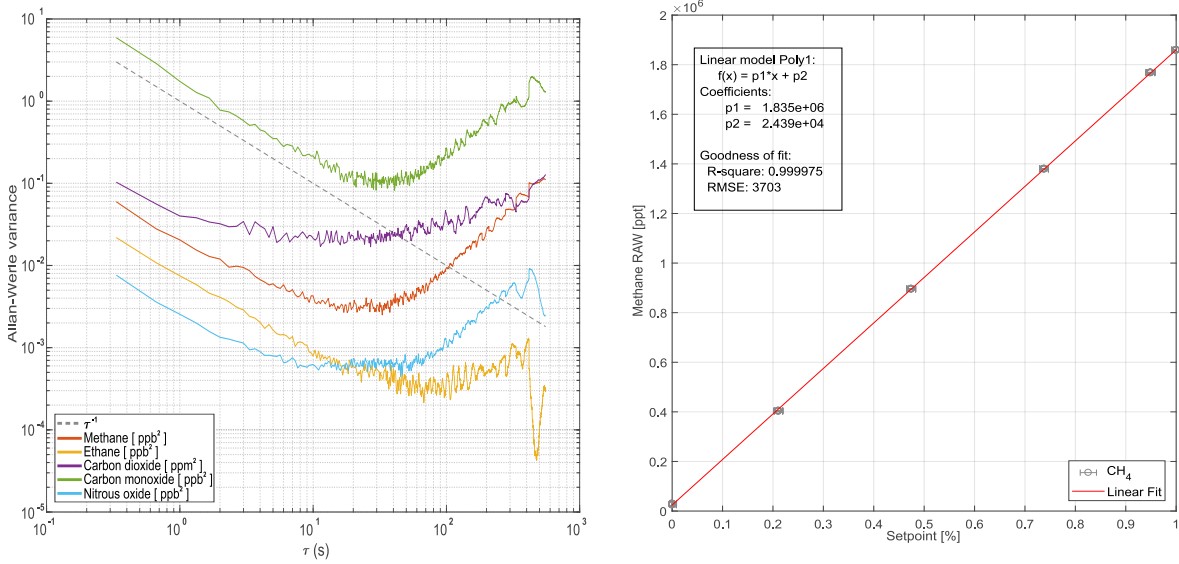

**Figure 7.** Allan-Werle variance for all measured chemical species during ground-based operation (left panel). The right panel demonstrates linearity for methane is within achievable error bounds during ground-based operation using the online calibration gas mixing system from Section 2.3.

Figure 7 shows the Allan-Werle variance for common averaging times $\tau$ for the individual trace gases monitored. For most species averaging up to 20 s will decrease the standard deviation of most of the signals, before deteriorating effects (i.e. drift) occur. Figure 7 also addresses retrieved mole fraction linearity. Linearity checks have been carried out for all species using the calibration system described in Section 2.3. All retrieved species are linear within the achievable controlled mass flow uncertainties from Section 2.3. $CH_4$ is used in Figure 7 for demonstration purposes.

Typical shift parameters (as introduced in Sect. 3) for ground-based operation are depicted in Figure 8 for the $CH_4$-$H_2O$ and $CO_2$-CO-$N_2O$ fit windows. These shift parameters can be considered as a tracer for instrument stability for both lasers. Overall spectral stability is in the range of $\pm 10^{-3}$ cm$^{-1}$. The regular short-timed spikes with a period of $\sim 5$ min result from switching from sample to calibration gas and vice versa. Apart from these well-timed spikes and expected low-frequency instability (due to thermal changes) on the lasers spectral output, high-frequency shifts are evident, including discontinuities. The source of these discontinuities remains unclear. They could be introduced by the software based frequency lock mechanism, by instabilities of the laser itself or by timing changes in the sampling. Software based frequency lock refers to a controller regulating the laser temperature to compensate for drifts using the spectral shift as the controller input and the current to the Peltiers as controller output. The controller itself is implemented in software on the data analysis computer. The shape of the individual shifts match and so does their trend over time, which is a good indicator for a stable tuning rate during ground-based operation.

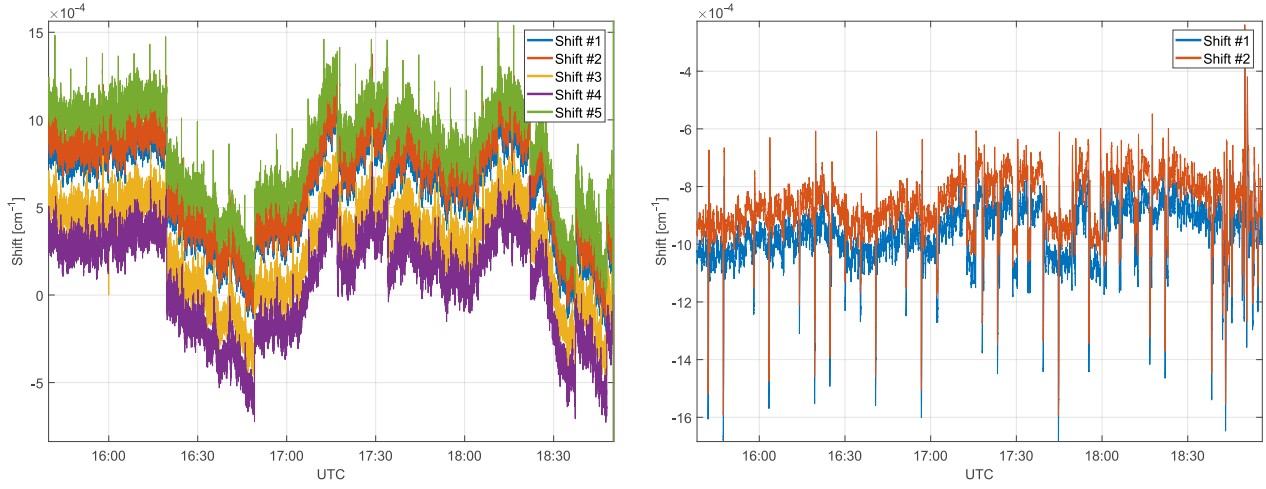

**Figure 8.** Spectral shifts for the $CH_4$-$H_2O$ fit window (left) and the $CO_2$-CO-$N_2O$ fit window (right). Spectral stability during ground-based operation is in the range of $\pm 10^{-3}$ cm$^{-1}$.

## 5 Airborne instrument performance aboard NASA WFFs C-130

The instrument was successfully operated during 18 research flights aboard NASA Wallops Flight Facility's C-130 within the framework of the ACT-America fall 2017 field campaign. Other instrumentation in the ACT-America payload provided an excellent opportunity for instrument inter-comparison. In situ $CH_4$, $CO_2$, and CO were measured using a Picarro G2401-m

5  cavity ring-down spectrometer, and in situ $CO_2$, $CH_4$, and $H_2O$ (g) were measured using a Picarro G2301-m cavity ring-down analyzer. Both cavity ring-down instruments are anchored to WMO X2007 for $CO_2$ (Zhao and Tans, 2006), WMO X2004A for $CH_4$ (Dlugokencky et al., 2005) and WMO X2014A for CO (Baer et al., 2002). Precise $C_2H_6$ measurements were obtained by periodic flask samples by NOAA ESRL. Three onboard lidars, and in situ sensors measuring the meteorological state variables - winds, temperature, pressure and water vapor - completed the C130s instrument suite. Here we present data from a

10  typical flight (10/03/2017) to demonstrate the airborne instrument performance through inter-comparison with well-established measurement techniques: the cavity ring-down greenhouse gas analyzers and flask samples.

As depicted in Figure 9, the flight starts off from the eastern U.S. (Wallops Flight Facility, Virginia). A high-altitude transect to West-Virginia is followed by two low-altitude legs downwind and upwind of parts of the Marcellus shale area: a large shale gas extraction region. The transects between the two low-altitude legs are flown at high altitude to facilitate nadir lidar

15  observation, with two en route descents and ascents near the center. Figure 10 depicts dry-air mole fractions for $CH_4$, $C_2H_6$ and $H_2O$ measured by the different instruments during the 5-hour flight. This figure provides evidence, that the QCLS and CRDS methane data agree to within 1.4 ppb ($1\sigma$) over the entire flight. QCLS and flask methane data agree to within 3.9 ppb ($1\sigma$). It should be noted, that care must be taken when interpreting the differences between slow flask samples and fast in situ measurements for high-variability flight segments. The center panel of Figure 10 depicts the QCLS-retrieved $C_2H_6$ data

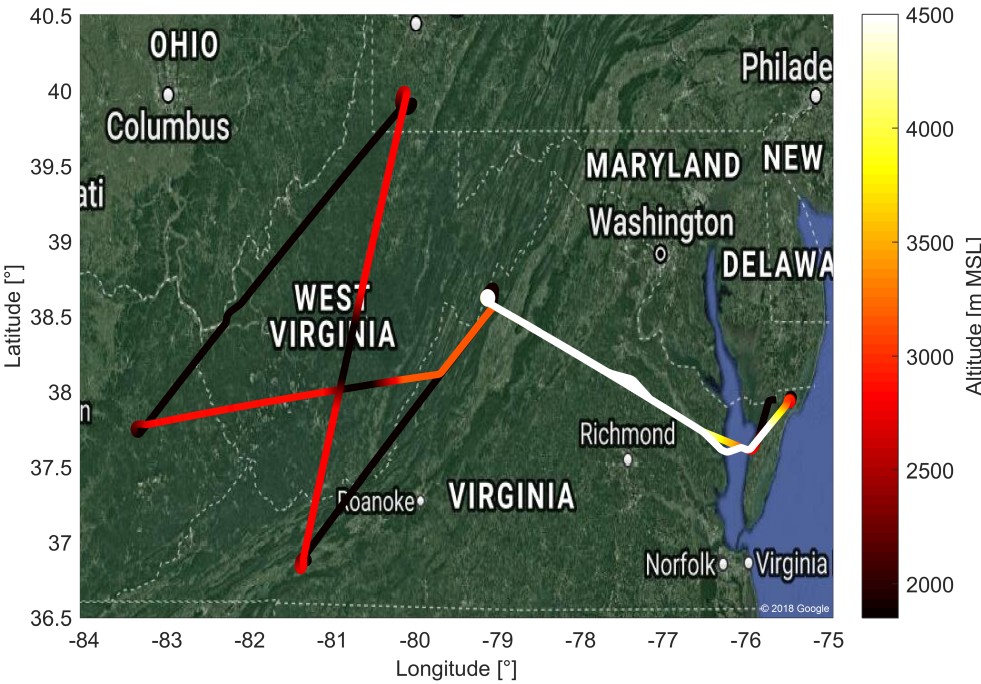

**Figure 9.** A typical flight during ACT-America. This figure shows the flight pattern for Oct. 3, 2017 with color coded altitude. The flight includes two low-altitude ($\approx 300$ m AGL) legs downwind and upwind of parts of the Marcellus shale area. High-altitude transects between the two low-altitude legs include two en route descents and ascents in West Virginia. Fair weather conditions and light southerly winds were present throughout the flight domain.

superimposed with flask measurements. Here the QCLS-retrieved ethane data matches the flask measurements (blue dots) to within $0.4$ ppb ($1\sigma$). Unlike the QCLS, cavity ring-down and flask data are both sampled through an upstream dryer. These were computed by interpolating QCLS data to the flask end fill times. The lowermost panel of Figure 10 provides water vapor mole fractions obtained from an on-board dewpoint hygrometer, from the G2301-m analyzer and from the QCLS. The QCLS

5 water vapor data is used to correct for water vapor effects during the retrieval of dry-air mole fractions from the QCLS raw spectra as described in Sect. 3.1. By taking a closer look on the upper two panels, the benefit of simultaneously measuring several species can be readily identified. Figure 10 shows enhanced $CH_4$ without coinciding $C_2H_6$ enhancements for the first low-altitude leg. For the second low-altitude leg above the Marcellus area, however, concurrent $CH_4$ and $C_2H_6$ enhancements suggest that natural gas is the dominant source.

10 Time series for the species $N_2O$, CO and $CO_2$ are shown in Figure 11. The data obtained are comparable within instrument uncertainties. The $N_2O$ time series matches available flask data to within $\pm 1.1$ ppb. The $CO_2$ absorption is retrieved from a molecular transition of the $^{13}C^{16}O_2$ carbon dioxide isotopologue and scaled with its natural abundance of approximately $1.1$ % (Gordon et al., 2017) to report total $CO_2$. Despite the much lower abundance compared to $^{12}C^{16}O_2$ the QCLS-retrieved

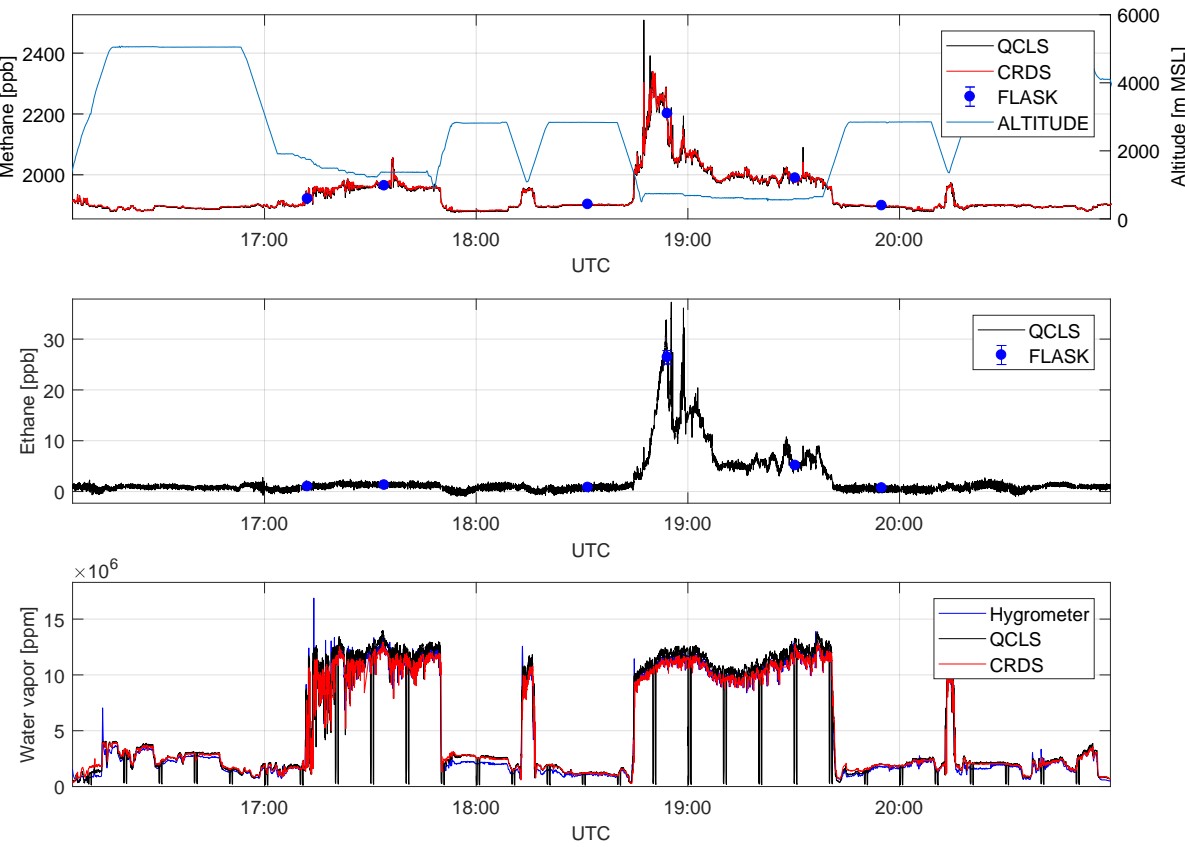

**Figure 10.** A direct comparison between dry-air mole fractions retrieved from different measurement techniques for a complete flight on Oct. 3, 2017. Depicted are methane (uppermost panel), ethane (center panel) and water vapor (lowermost panel) mole fractions. QCLS-retrieved methane data matches with CRDS and flask data to within 1.4 ppb ($1\sigma$) and 3.9 ppb ($1\sigma$), respectively, after correcting for a constant bias. QCLS-retrieved ethane data agrees with flask data to within 0.4 ppb ($1\sigma$). Water vapor sensed by an on-board dewpoint hygrometer does differ from the CRDS and QCLS data.

$CO_2$ data coincides with cavity ring-down data to within $\pm 0.6$ ppm ($1\sigma$) after correcting for a constant bias (see below). QCLS-retrieved CO mole fractions (center panel) agree with CRDS-retrieved data to within $\pm 5$ ppb ($1\sigma$). Figure 11 suggests that in-flight precision depends on whether flying within the planetary boundary layer or above it. This is due to aircraft vibration excited by running engines and turbulence propagating into the instrument optics inducing slight changes in optical
5  alignment and enhanced natural variability in the planetary boundary layer. We identified temperature fluctuations within $\sim 0.3$ K, pressure changes of up to $\sim 200$ hPa and relative humidity changes of up to 35 % in the instruments optical compartment during this flight.

Typical in-flight precision figures based on ambient measurements at stable conditions for both regimes (standard deviation for 1s averaging) are summarized in Table 3. Total measurement uncertainty can be estimated from the uncertainty of the working
10  standards, the uncertainty of calibration sequence evaluation, the uncertainty introduced by the $H_2O$ correction, the precision

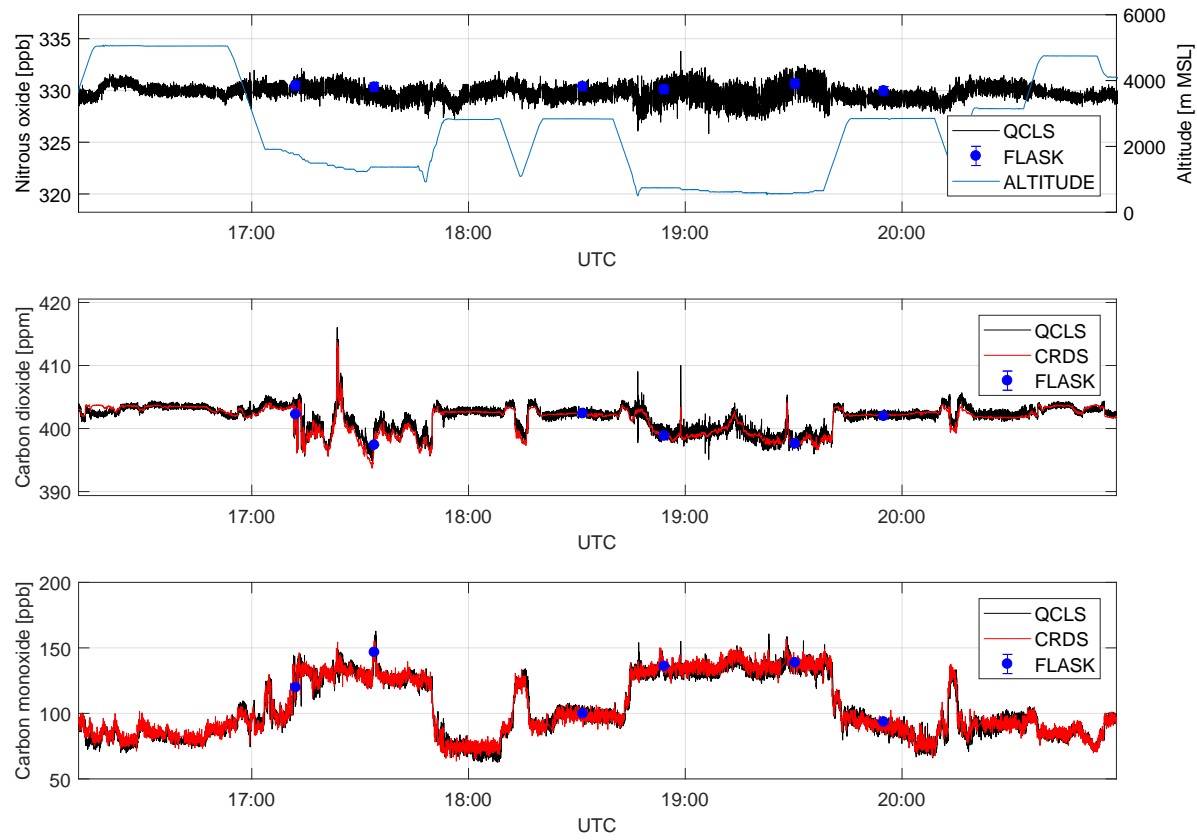

**Figure 11.** Dry-air mole fractions retrieved from different measurement techniques for a complete flight on Oct. 3, 2017. Depicted are nitrous oxide (uppermost panel), carbon dioxide (center panel) and carbon monoxide (lowermost panel) mole fractions.

of the instrument and errors due to drift. We found a bias constant for the whole measurement series of $\sim +2$ ppb for $CH_4$ and $\sim +10$ ppm for $CO_2$ between the QCLS and CRDS/FLASK datasets. This constant bias has been corrected for. The origin of the biases is not yet fully understood. It was suggested that water vapor correction could have an impact on this. The reason for this assumption is that the calibration standards are always dry, whereas sampled air is not dried before entering the sample

5 cell. Correlation plots however show no signficant influence of water vapor on the residuals between the dry-air-sampling CRDS and the QCLS. It is therefore very unlikely that the water vapor correction is the source of the large bias in $CO_2$. Instead we identified the difference in isotopic composition of the calibration standard versus sampled atmospheric air as the most probable cause. In this study we used working standards of synthetic nature from *Air Liquide*. Usually these are produced with $CO_2$ from natural gas & oil combustion processes. We determined the $CH_4$ and $CO_2$ values of each working standard

10 gas cylinder using a NOAA-anchored (Cert.-Nr. CB11361) Picarro G-1301m. This has the drawback that we do not know the isotopic composition of our working standards as its impact had been considered negligible, e.g. Chen et al. (2010). We learned during development of JFIT, that the instrument is using a $^{13}C^{16}O_2$ line to derive ambient $CO_2$. We estimate the required

**Table 3.** Typical in-flight performance including contributions to overall uncertainty. The total measurement uncertainty at 1 s temporal resolution is given by the quadrature sum of the individual contributors. Due to the lack of the appropriate NOAA standards during the deployment, the uncertainties in $C_2H_6$, CO and $N_2O$ include combined uncertainties from concurrently measuring instruments (CRDS & Flasks). The total uncertainties stated for these species do therefore not reflect the intrinsic uncertainties of the instrument, but worst-case values, that may be better given the availability of appropriate standards. The WMO compatibility goals for Global Atmosphere Watch network compatibility among laboratories and central facilities have been added for completeness.

| Chemical species | $H_2O$ | $CH_4$ | $C_2H_6$ | $CO_2$ | CO | $N_2O$ |
|---|---|---|---|---|---|---|
| Precision $1s - 1\sigma$ (within PBL) | 16.2 ppm | 740 ppt | 205 ppt | 460 ppb | 2.2 ppb | 439 ppt |
| Precision $1s - 1\sigma$ (above PBL) | 2.5 ppm | 300 ppt | 146 ppt | 182 ppb | 1.4 ppb | 208 ppt |
| Working standard reproducibility ($1\sigma$) | — | 0.03 ppb | — | 0.1 ppm | — | — |
| Compared instrument uncertainty ($1\sigma$) | — | — | 1.5 ppb | — | 5.0 ppb | 0.4 ppb |
| Measurement calibration ($1\sigma$) | — | 1.5 ppb | 0.5 ppb | 0.9 ppm | 4.4 ppb | 0.5 ppb |
| $H_2O$ correction ($1\sigma$) | — | 0.8 ppb | 0.1 ppb | 0.2 ppm | 0.2 ppb | 0.1 ppb |
| WMO compatibility goal | — | 2.0 ppb | — | 0.1 ppm | 2.0 ppb | 0.1 ppb |
| Total 1s-1$\sigma$ uncertainty | — | 1.85 ppb | 1.6 ppb | 1.0 ppm | 7.0 ppb | 0.8 ppb |

isotopic composition that could explain the large bias of 10 ppm (see Supplement Section 3) in $CO_2$ to be 98.447 % primary isotopologue and 1.079 % secondary isotopologue or $\delta^{13}C = -19.6\,‰$ which seems reasonable according to B. Coplen et al. (2002). Since we are reporting retrieved mole fractions relative to the WMO scale, only the working standard reproducibility contributes to the total uncertainty of $CH_4$. Comparability of $CO_2$ is difficult to assess here because of the unknown isotopic composition in our working standards. Uncertainty in the other measured species is taken from the ACT-America dataset to allow for WMO traceability. Due to the lack of the appropriate NOAA standards during the deployment, the uncertainties in $C_2H_6$, CO and $N_2O$ include uncertainties reported in the ACT-America dataset from concurrently measuring instruments (CRDS & Flasks). The total uncertainties stated for these species do therefore not reflect the intrinsic uncertainties of the instrument, but worst-case values, that may be better given the availability of appropriate standards in future deployments. The uncertainty of calibration sequence evaluation (see Section 2.3) is estimated with the double of the measurement precision and the uncertainty introduced by the $H_2O$ correction is estimated from Eq. (1) using an assumed relative error on retrieved water vapor of 2 %. Errors originating from instrument drift are considered negligible due to our frequent calibration strategy (see Section 2.3). The total uncertainty is given by the quadrature sum of the individual contributors, listed in Table 3. Table 3 further includes the WMO compatibility goals for Global Atmosphere Watch (GAW) network compatibility among laboratories and central facilities. Precision/uncertainty figures given in Table 3 can be compared to 2s-1$\sigma$ PICARRO G2401-m airborne precision/uncertainty estimates based on ambient measurements at stable conditions of 0.3/2 ppb, 0.02/0.1 ppm and 2.0/5 ppb for $CH_4$, $CO_2$ and CO, respectively.

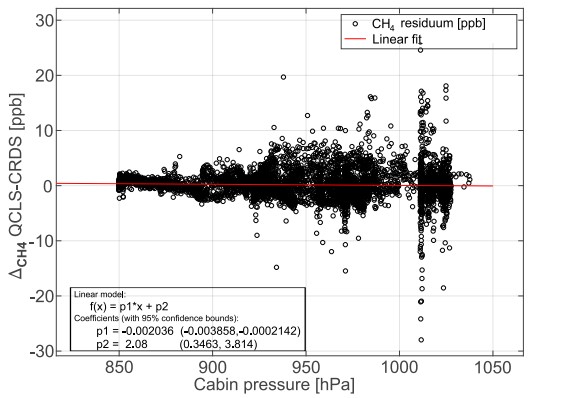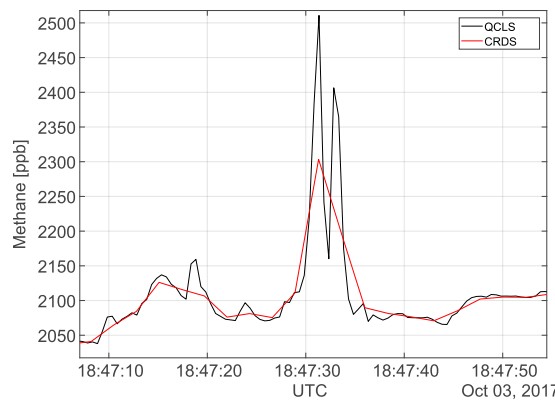

**Figure 12.** The left panel shows the cabin pressure dependence for a typical flight on Oct. 3, 2017. The large cabin pressure dependence in excess of $0.3\,ppb\,\mathrm{hPa}^{-1}$ reported by Pitt et al. (2016) has effectively been minimized using the calibration strategy from Sect. 2.3. The right hand side panel shows a temporal zoom on the $CH_4$ mole fractions at 18:47 UTC to emphasize the benefit of high-frequency measurements.

A severe cabin pressure dependence in excess of $0.3\,\mathrm{ppb}\,\mathrm{hPa}^{-1}$ in $CH_4$ mole fraction has been previously reported for airborne TILDAS instrumentation (Pitt et al., 2016). This instrumentation however physically differs from the one reported in this study. It is not possible to accurately compare the dependencies of one instrument relative to another since many factors/quantities involved are instrument-specific, e.g. the open-path length, the positioning and properties of optical elements, like windows,

mirrors, etc., the stiffness and thermal expansion coefficient of the employed optical stands. We were nevertheless able to effectively minimize cabin pressure dependencies during operation of the QCLS instrument aboard the C130 using the calibration strategy from Sect. 2.3. This required a total calibration gas amount of $\sim 3.5\,\mathrm{m}^3$ (excluding zero air) for the 18 research flights. Figure 12 (left panel) shows the difference in $CH_4$ dry-air mole fraction reported by the QCLS and the CRDS as a function of cabin pressure during the research flight described above. The large scatter results from different sampling patterns among the

two instruments, hindering a one-to-one comparison of the QCLS measurements with the CRDS. While the QCLS samples continuously with a frequency of 2 Hz (1.5 kHz sweep frequency), the CRDS samples with a frequency of 0.5 Hz one species after the other. For $CH_4$, for example, the CRDS uses the first $0.5$ s of the $2$ s sampling time, implying that, for the later $1.5$ s, the CRDS is insensitive to $CH_4$. Therefore, it is difficult to mimic the cavity ring-down sampling by averaging the QCLS data as it would be required for a one-to-one comparison. Instead we decided to linearly interpolate QCLS data to the CRDS

timescale. The fast response time of the QCLS instrument allows for better sampling of spatially narrow plumes, as can be seen from the right hand side panel in Figure 12. This panel zooms in on a relevant portion of the methane data from Figure 10 and demonstrates that two mutually-separated plumes can be identified from the high frequency QCLS data at 18:47 UTC, where only a single enhancement can be seen from cavity ring-down data. Furthermore, absolute enhancement and area beneath the peak(s) differ for the two instruments, due to the different sampling patterns. Figure 13 compares the QCLS mole fractions to

the cavity ring-down instrument and to the flask samples after correcting for a bias constant for the whole measurement series. The upper panels show differences in retrieved mole fractions between the QCLS and the cavity ring-down instrument for the

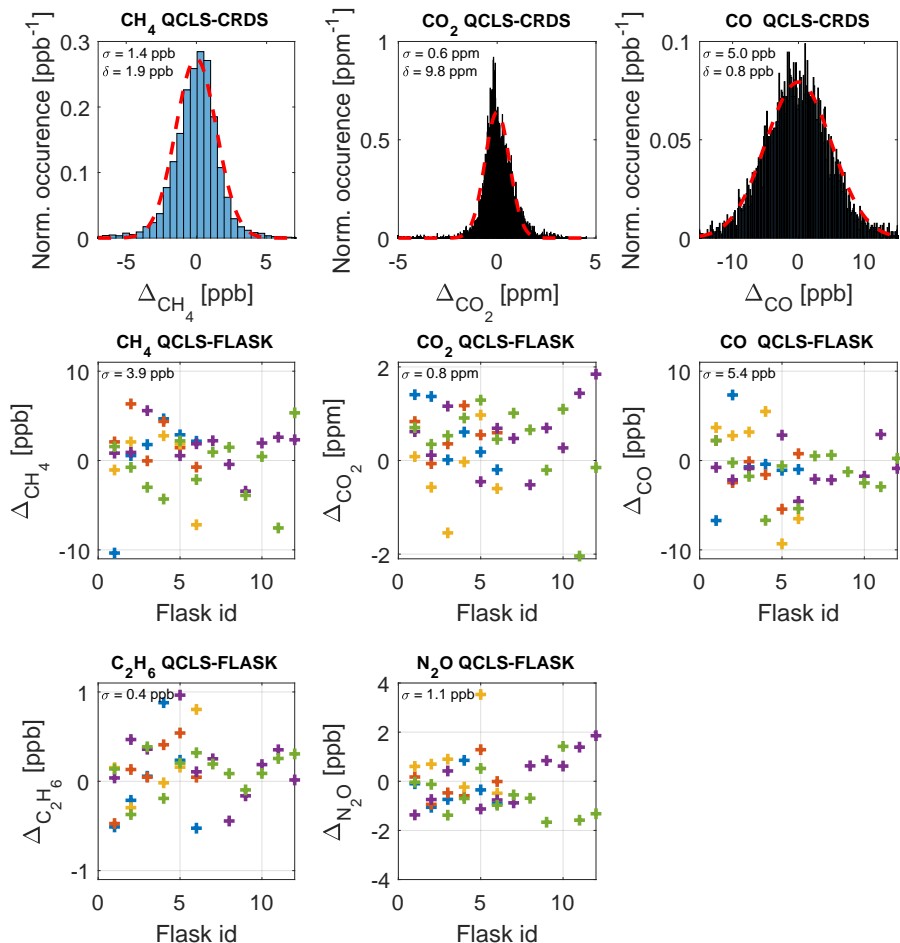

**Figure 13.** Comparison of QCLS derived mole fractions to well-established in-flight cavity ring-down data and flask samples after correcting for a bias ($\delta$) constant for the whole measurement series including standard deviations $\sigma$. Interpretation of the errors against flask samples is difficult for high-variability flight segments, due to the large flask sampling time. The residual plots show color-coded data from 5 typical flights on 10/03/2017 (blue), 10/11/2017 (red), 10/14/2017 (yellow), 10/18/2017 (violet) and 10/20/2017 (green).

flight on Oct. 3, 2017, exhibiting a near normal distribution. This hints towards residuals originating from random processes, i.e. noise. Despite the different sampling time and pattern, the measurements exhibit a compatibility to the calibrated cavity ring-down observations of $1.4$ ppb in $CH_4$, $0.6$ ppm in $CO_2$ and $5.0$ ppb in CO. Although interpretation of the differences to flask samples is difficult for high-variability flight segments, the lower panels of Figure 13 show a good agreement for five typical flights (10/03/2017, 10/11/2017, 10/14/2017, 10/18/2017, 10/20/2017) during the ACT-America campaign. The relative deviations are in good agreement with the QCLS-CRDS data, except for $CH_4$, where QCLS-CRDS compatibility ($1.4$ ppb) differs from the QCLS-FLASK compatibility of $3.9$ ppb. This could be related to the different sampling times between the QCLS and flask samples.

## 6 Conclusions

We adapted the commercially-available QCL/ICL based *Dual Laser Trace Gas Monitor* from *AERODYNE RESEARCH INC., Billerica, USA* for airborne flux estimation (e.g. via the mass-balance approach) and demonstrate successful operation for representative research flights aboard NASA Wallops Flight Facility's C-130 during the ACT-America field campaign in fall 2017. Known cabin-pressure dependencies (Gvakharia et al. (2018); Pitt et al. (2016)) on the retrieved mole fractions are effectively minimized using a frequent (5 to 10 min interval) two-point calibration approach obtained by flushing the sample cell with "zero" and "target" gases. This allows for a measurement duty cycle of $\geq 90$ % when operating at sample flow rates near 23 SLPM. A custom retrieval software has been developed to learn about possible error sources, mitigation possibilities of instrument dependencies and to be able to extend the instruments capabilities in the future. Apart from low frequency laser instability we identify high frequency "jumps" on the spectral axis, possibly due to the instruments frequency lock mechanism. In-flight performance has been assessed using data obtained during the research flight on the 3rd Oct. 2017 above the eastern US. We identify two precision regimes whether flying within the planetary boundary layer or above, due to aircraft vibration propagating into the instrument optics and related slight changes in optical alignment. Typical in-flight precision figures for boundary layer flights (standard deviation for 1s averaging) are 740 ppt, 205 ppt, 460 ppb, 2.2 ppb, 137 ppt, 16 ppm for $CH_4$, $C_2H_6$, $CO_2$, CO, $N_2O$ and $H_2O$ respectively. Precision figures improve to approximately the half for flights above the PBL. We estimate a total measurement uncertainty of 1.85 ppb, 1.6 ppb, 1.0 ppm, 7.0 ppb and 0.8 ppb in $CH_4$, $C_2H_6$, $CO_2$, CO and $N_2O$, respectively. We demonstrate QCLS comparisons to concurrent flask sample and cavity-ringdown measurements within combined measurement uncertainty for all targeted species. The instrument retrieves carbon dioxide mole fractions via a $^{13}C^{16}O_2$ absorption line. We find that precise knowledge of the $\delta^{13}C$ of the working standards and the sampled air is needed to enhance $CO_2$ compatibility when operating on the 2227.604 $cm^{-1}$ $^{13}C^{16}O_2$ absorption line.

*Code and data availability.* Software code and data are available from the authors upon request.

*Competing interests.* The authors are not aware of any competing interests.

*Acknowledgements.* We thank DLR VO-R for funding the young investigator research group "Greenhouse Gases". We also acknowledge funding from BMBF under project "AIRSPACE" (Grant-no. FKZ01LK170). The Atmospheric Carbon and Transport (ACT) - America project is a NASA Earth Venture Suborbital 2 project funded by NASA's Earth Science Division (Grant NNX15AG76G to Penn State). Aircraft operations and U.S. investigators were funded via the ACT-America project. We greatly appreciate continuous support from Hans Schlager and Markus Rapp from DLR. We further thank the GLORIA team, especially Christof Piesch and Hans Nordmeyer at the Karlsruhe Institute of Technology (KIT) for enabling us pressure testing the rack-mounted QCLS in a climate chamber. We also thank Paul Stock and Monika Scheibe from DLR and Martin Nowicki from NASA WFF for engineering support. Furthermore we would like to thank everyone involved during the ACT-America field campaigns for their relentless dedication and the helpful discussions, especially Alan Fried, Bing Lin, John Nowak, Linda D. Thompson, Charles E. Juenger. A special thanks to Mark Zahniser and Dave Nelson from Aerodyne Inc. for their valuable support whenever help was needed.

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
