# Peer review of "Adaptation and performance assessment of a Quantum / Interband Cascade Laser Spectrometer for simultaneous airborne in situ observation of CH4, C2H6, CO2, CO and N2O"

_Atmospheric Measurement Techniques, 2018_

## Referee Comment (RC1) · A. Fried (Referee) · 16 Oct 2018

This is an excellent paper representing very careful and well thought out procedures and analysis methods. The paper is very well written and the results are very sound. I particularly like the fact that two different precision regimes have been identified (within the PBL and above the PBL) and yield different results due to differences in alignment caused by aircraft vibrations. We often see this effect in our measurements and highlighting them here is a further illustration of the care devoted to the measurements presented. The one thing that should be added is a brief section indicating how the in-flight precisions were determined. Did the authors base this on the precision of zero air measurements or the precision of ambient measurements under stable conditions? In the case of the latter, ambient variability cannot be ruled out the in-flight precisions may be even better than indicated.

I recommend final publication after the following minor points are addressed. As you can see, these are all very minor and serve to clarify some of the discussion.

1. Introduction, Line 6: change the word "remain" to "have"

2. Introduction, Line 16: Since there have been extensive measurements of atmospheric gases well before QCLs and ICLs in the mid-IR using for example, liquid nitrogen cooled lead-salt diode lasers as well as other sources, a brief sentence giving a reference to some of this work should be included. One can cite numerous sources, but one convenient way (at the risk of being self-serving) would be to cite our text book chapter which has many of these references (Chapter 2: Infrared Absorption Spectroscopy by A. Fried and D. Richter, in the book Analytical Techniques for Atmospheric Measurement, edited by D.E. Heard, Blackwell Publishing, 2006).

3. Introduction, Line 20: Please change the wording "custom-built QCL..." to "custom-built difference frequency generation (DFG) absorption spectrometer"

4. Introduction, Line 27: Are you strictly referring to established cavity ring-down instruments here or are you referring to more generally IR absorption instruments? Since you mention the "described spectrometer", I think you should change "established cavity ring down" to "established IR spectrometers"

5. Page 4, Line 25: It would be useful to the reader to further elaborate on the meaning of "jeopardized nominal system startup". Do you simply mean the large in rush current to get the pump going is more than the airplane circuit breakers can take, or does this

AMTD
mean that this may cause damage to other parts of the instrument?

6. Figure 5: It would be very helpful to the reader to indicate the mixing ratios in the figure caption used in recording the various spectra.

7. Page 10, Line 2: Where is the weak CH4 line relative to the C2H6 line which is used for spectral shifting of the C2H6 feature?

8. Page 11: This is a very nice discussion of the various broadening parameters and how they are handled. However, this reviewer wonders how important actually including the self-broadening and water broadening are in the final fits since these are smaller by the fact the sampling pressure is 50-mb and the overall spectral stability is in the 10-3 cm-1 range? The air broadening at this pressure is only  $\sim$ 0.0035 cm-1 which is close to the spectral stability. Perhaps a brief mention of how the inclusion of self and water broadening changes the retrieved results should be included.

9. Figure 7 and Its Caption along with Page 13, Line 1: At the FLAIR (Field Laser Applications in Industry and Research) the Program Committee strongly recommended that references to "Allan Variances" should be denoted "Allan-Werle Variances" in honor of the late Peter Werle who adapted this concept to atmospheric measurements. Below I include a portion of the Program Committee's Obituary for Peter Werle and its recommendation (this need not be included in the final paper but is included here for your reference). Also, what mixing ratios were used in recording Fig. 7 (zero air or calibrated standard mixing ratios)?

In 1993 Peter and colleagues published a seminal paper in Applied Physics B in which they introduced the concept of Allan Variance applied to laser diode measurements of gas concentration time series. The principle of using Von Neumann's two-sample variance to describe the statistics of time series that exhibit power law spectra that are more dispersive than white noise (e.g., show a varying mean, or drift) was first made popular by D.W. Allan in 1966 for characterizing frequency standards. The importance of introducing this new tool to laser physicists and atmospheric scientists cannot be AMTD
overstated; it provides a wealth of new information in a mathematically formal and rigorous framework that would not otherwise be easily accessible. Today virtually every group in the world carrying out ultra-sensitive laser spectroscopic measurements employs the concepts introduced by Peter and colleagues. It is indeed rare to find a highquality paper discussing measurements without an Allan Plot. Perhaps such plots should in the future be referred to as "Allan-Werle" plots.

10. Page 15, Discussion of Fig. 10: In comparing flask and in situ measurements it should be mentioned that care must be exercised in that during times of rapidly changing ambient mixing ratios one may not get agreement between the slow flask samples and fast in situ measurements. Although this is obvious, it is worth mentioning here. I see this is discussed in the Fig. caption 13 but it is also worth mentioning here.

11. Page 18, in the discussions of cabin pressure dependence: The authors should mention that it is not possible to accurately compare the dependence of one instrument relative to another since many instrument-dependent and other factors come into play. For example, some of the dependence is due to the changing mixing ratios for the species under study in the open-air path. Additional dependencies result from movement of optical windows and other components and are instrument dependent. Also, we find that the rate of cabin pressure change is an important factor, and this is specific to the particular aircraft and the flight pattern employed. Hence, the left side of Fig. 12 may not tell the whole story. We find that the delta Pcabin/delta time comes into play between zero acquisitions, and I would expect the same thing here. Perhaps a comment on this should be mentioned.

The following does not need to be addressed in this paper but is for your informational purposes only. Given your zeroing/calibration every 5 to 10-minutes, you may find that this is too infrequent for aircraft undergoing large rates of cabin pressure changes like the NASA DC-8 where the cabin pressure can change as much as many torr/second. The ultimate solution would be to build a pressure-stabilized enclosure.

---

## Referee Comment (RC2) · Anonymous Referee #4 · 30 Oct 2018

The paper by Kostinek et al. presents ground-based and in-flight performance assessments of a commercial trace gas analyzer (TILDAS, Aerodyne Research Inc.) after its adaptation for airborne operation. The subject is highly topical and targets a key issue that every scientist is facing when taking decision on analyzer selection. This can even be critical considering the stringent place and measurement-time limitations of flight campaigns. Here, the author's choice is on a dual-laser direct absorption spectrometer with multi-species (i.e., five compounds) detection capabilities deploying

state-of-the-art mid-IR laser sources (both QCL and ICL). The in-flight intercomparison with CRDS-based instruments and flask samples is an important element of the manuscript that can be of interest for the community involved in airborne measurements. The manuscript is well written, and I recommend publication after addressing some comments and changes listed below.

General:

The title should better reflect the content of the manuscript. Given that for the measurements the spectrometer is equally using both QCL and ICL devices, this should be weighted the same. Furthermore, the instrument was mainly adopted and not modified for airborne operation. Thus, my suggestion is to write: "Adaptation and performance assessment of a dual-laser mid-IR direct absorption spectrometer for ..."

The abstract should focus on summarizing briefly the highlights and findings of the presented work. Thus, I suggest starting directly with L7, "Here we demonstrate..."

Although the introduction contains a brief hint about the large number of available meas-urement techniques for airborne atmospheric measurements, it is unfortunate that the authors completely refrain to motivate their choice for a particular analyzer. Clearly there are some benefits of having multi-species capabilities at slightly higher sampling rate, but how this compensates for the obvious limitations, such as cabin pressure dependence, frequent high-flow calibration requirements, tedious post-processing of the raw spectral data, high power and calibration gas consumption, etc.? A more elaborate discussion on ad-vantages/disadvantages of the chosen approach would significantly improve the manuscript.

In the same context, it is also not obvious in the present form, why the authors decided for an extensive calibration scheme and additional data post-processing instead of developing a purged and sealed enclosure around the instrument. Apparently, most of the relevant drifts or biases are due to ambient air (H2O mainly) absorptions outside the multi-pass cell.

Considering that the spectral retrieval software of the manufacturer has been around for more than 20 years, supporting a large number of custom-built instruments applied in a wide range of applications, one would assume that the software experienced a continuous development and incorporates many fine-tuning/customizing features in order to optimize also the fitting process. Therefore, it is highly interesting and valuable if the custom retrieval software (JFIT) significantly improves the performance. This must, however, be more clearly documented by a side-by-side comparison of the results of both software packages. Especially, the additional shift parameters introduced by JFIT and its co-allocation to various segments of the spectral window seems rather subjective and should be quantified in terms of performance improvements. The authors refer to Fig.8 (p13) to show that the tuning rate of the lasers is stable, which seems to be in contradiction with the many shift parameters introduced in JFIT.

Similarly, the discussion of the water vapor correction should contain further details about the observed biases and drifts in the whole range of water concentrations experienced during flights. Measuring water concentration at absolute level is challenging, so the authors should show the observed correlation between generated humidity and spectroscopically measured water mixing ratios. Also, some discussion is required to make clear how the additionally in-troduced broadening effect improves the measurements, and compare this to the impact of the significant spectral instability (10-3 cm-1) and potential temperature fluctuations of the sampled gas and of the cabin during flights.

Since the instrument is a unique platform, where two different mid-IR laser sources are oper-ated side-by-side, a more detailed comparison of performances and noise characteristics of QCL/ICL would certainly be an added value to the manuscript.

Specific:

Pg2, L27: need more clarification what is meant by sequential and truly concurrent sensing. Otherwise, there should be a short note about the importance/benefit of measuring at 0.5 Hz instead of 2 Hz.

Pg2, L25: a reference to the paper at this stage is enough; especially, that Section 2 starts with the same information.

Pg2, L29-31: How to interpret these cited works? In the present context, they give the impres-sion that there is no open question regarding the suitability of QCLS for airborne measure-ments.

Pg2, L32: the main objective of the paper is missing. What is the final goal of this investigation? Which measurement data and for what purpose are they going to be used? Is the data quality adequate to answer the research questions?

Pg3, Sect 2.1: this section needs some re-work, e.g. statement like "optics compart-ment con-tains all optical elements" is redundant, while Laser#1 and #2 without clear definition has no sense. I suggest giving the driving specifications of the lasers (current and temperature) as well as their optical power output. Specify the exact detector type.

Pg4, L1: obviously there are many hundreds of reflections within the multi-pass cell, which leads to significant decrease of optical power of the laser beam. What is the reflectivity of the mirrors and how much is the out-coupled ICL intensity?

Pg4, L3: I doubt that the laser devices are directly coupled to the Peltiers. There should be a buffer heat-sink between.

Pg8, L15: the relative frequency changes seem to be the same, which is also illustrated by Fig8. So what is the real benefit for using five different shift parameters?

Pg9, L1: how large were the temperature fluctuations within the optical compartment? What was their effect on the spectral retrieval?

Pg9, Fig 9: specify the averaging time of the spectra.

Pg9, Fig 9: where is the CH4 line in the C2H6 fit-window? What is causing the strong bias in the residual in the CH4 fit-window?

Pg10, L7: a short clarification should be added why the authors chose this difficult spectral window? The range around 2224.5 cm-1 would, e.g. contain all the species with significantly less spectral interference. The ambition of getting the CO2 along with CO and N2O introduces severe compromises in the achievable spectral sensitivity and selectivity. Adding the fact that the selected CO2 line is not even the main isotopologue and seems to have large systematic bias, I seriously doubt whether this compromise is worthwhile.

Pg11, L9: indicate the precision and accuracy of the generated water vapor mixing ratios. What about hysteresis effects, i.e. humidifying vs. drying cycle?

Pg12, L4: How well can the results obtained at 1 SLPM transferred to the 23 SLPM operation regim? What about simply using an empirical correction factor on the retrieved mixing ratios instead of introducing the broadening coefficient in the fitting procedure?

Pg12, Fig.7: it seems that the plot shows the deviation instead of variance. The Allan deviation plot indicates that the instrument drifts already after 30 s even though operated under ground-based conditions. During flight, pressure and temperature variations, as well as mechanical vibrations tend to impair the performance of the instrument at even shorter time-scales. Considering the long-path of the optical cell, I wonder whether the authors did observe any correlated noise behavior when changing gas flow through the cell from 1 to 23 SLPM? As such, it would be useful to see the distribution diagram (or at least to give quantitative estimates of their spread) of the calibration gas measurements during flights.

Pg13, L9: what is meant by software based frequency lock mechanism?

Pg13, Fig8: what is the influence of the sudden frequency shift discontinuities on the retrieved mixing ratios?

Pg17, L4: what would be the required isotopic composition of such a CO2?

Pg17, L8: give an estimate of the overall calibration gas consumption for the 18 flights

and shortly discuss options for optimization.

Pg19, L10: as mentioned earlier, it would be useful in this context to show the distribution di-agram of the calibration gas measurements performed at every 10 min interval and repre-senting about 10% of the measurement time.

Pg19, L17: it is somehow unclear what applies: in the previous section (pg18, L2) the authors claim that they were unable to reproduce the cabin pressure dependence, but in the conclusion is argued that the known cabin-pressure dependencies are effectively minimized by frequent two-point calibration.

Pg19, L24: was the frequency lock mechanism active during flight operation only? Do the fre-quency-"jumps" correlate with laser heat-sink temperature changes?

Pg20, L2: Having an uncertainty of 1 ppm and systematic bias of 10 ppm on the CO2 retrieval, projecting towards isotope ratio measurement is quite steep.

Technical corrections:

Pg1, L12: "truly" is not a proper attribute for simultaneous. Remove it.

Pg2, L15: check reference, because Santoni et al. used QCLS instead of CRDS

Pg2, L20: as above, Richter et al. used DFG instead of QCL

Pg3, L16: here and across the manuscript add space between value and unit. Also the chemical formula should be always printed in Roman (upright) type (see e.g. IUPAC Green Book).

Pg4, L3: avoid using laser diode when referring to ICL/QCL devices.

Pg8, L4: replace "micro" by "fit" window.

Pg21, references: check for typos and completeness, e.g. at L10, L13, L22, etc.

---

## Referee Comment (RC3) · Anonymous Referee #2 · 9 Nov 2018

Atmos. Meas. Tech. Discuss., https://doi.org/10.5194/amt-2018-312

**Review of the submitted article:**

**Modification, Characterization and Evaluation of a Quantum/Interband Cascade Laser Spectrometer for simultaneous airborne in situ observation of $CH_4$, $C_2H_6$, $CO_2$, CO and $N_2O$**

Julian Kostinek, Anke Roiger, Kenneth J. Davis, Colm Sweeney, Joshua P. DiGangi, Yonghoon Choi, Bianca Baier, Frank Hase, Jochen Groß, Maximilian Eckl, Theresa Klausner, and André Butz

General comments:

This paper describes the performances of an analyser of the major greenhouse gases in air on board of an aircraft. Details are provided on the analyser hardware, the analytical software and the calibration method, followed by an evaluation of the performances by comparison with other analysers present during the same flight. While the analyser itself is not new and was already described in a previous paper (McManus 2011), in this work the number of analysed compounds was extended, the in situ performances were looked at more deeply, and the calibration method was improved. The paper is generally well written, well-structured, clear, and provides lots of details on the instruments and methods. However the section on performance evaluation needs some more work, both in its content and format. I therefore recommend a minor revision before the paper can be published in AMT.

Comments on the terminology

- Units to be written in plain (not italic) format

- The format to display a value with its unit is "value−space−unit" for example "204*m*" on page 4 should be written "204 m"

- "mixing ratio" to be replaced by "amount fraction", expressed in mol mol$^{-1}$ (nmol mol$^{-1}$ for ppb, µmol mol$^{-1}$ for ppm).

- Names of molecules to be written in plain (not italic) format.

- Allan deviation seems to be confused with Allan variance. When values are reported in the same unit as the concentrations, this should be a deviation. Please check the correct usage over the document.

Specific comments by section:

Section 2.1: the text describes two sealed cells containing $CH_4$ and $N_2O$. Where are they on Figure 1? Please indicate the purity of the gas and its pressure.

Section 2.3: please provide more information on the calibration mixtures. In particular NOAA standards are all identified within NOAA database and you could just provide their reference to allow the users looking at all values measured by NOAA. At least please indicate the nominal amount fractions, their uncertainties, and the isotopic composition for $CO_2$. This last value is of importance as you noticed a bias between the $CO_2$ amount fractions measured with your instrument and those measured by the PICARRO.

Section 4, ground−based performance: the reported Allan deviations seem a bit large. Compared to McManus 2011 on $CO_2$ for example, a factor 10 is noted. Please consider revising the statement that "values are in good agreement with the values reported by Aerodyne" and/or provides further support. Is there an effect of the calibration system described in 2.3, which is said to be used to check the stability and the linearity?

Section 5: while the traceability of measurements with the QCL is clear (calibration with NOAA standards), nothing is indicated regarding the PICARRO. This is needed to fully understand the origin of biases. It seems that an anchored to NOAA is assumed, but this deserves further details (which standards? How many calibration steps? Isotopic composition?).

When both instruments are compared, it would be more useful to express the difference in amount fraction, both in the text and in the graphs. This should then be compared with their uncertainties, not taking into account common sources of uncertainties such as NOAA uncertainty if all amount fractions are expressed on the same scale. Going beyond this, some consideration on how this compares with the Data Quality Objectives set by WMO would be of interest.

The treatment of the constant bias found between the AERODYNE and the PICARRO analysers needs to be improved. Are both analysers calibrated directly with NOAA standards? How different are the calibration gases? It would be valuable to estimate the bias one could expect from the isotopic difference, as done for example in the paper of Chen et al. (Atmos. Meas. Tech., 3, 375–386, 2010), and compare with the observed bias. Indeed, an observed bias of 10 µmol mol$^{-1}$ seems very large.

Section 5, uncertainties: it is not so common to see combined uncertainties considered in such measurements, and the effort of the authors is certainly valuable. However some consideration on how these values compare with other instruments would be required. Is the calibration procedure specific to this instrument? Does this imply a larger uncertainty than for others? Would you say this instrument has comparable precisions than others?

Section 5, discussion on instruments precisions: Allan deviations (not variance) were measured before the flight and during the flight. It is not very clear how those values compare. One would expect the lowest values during ground−based measurements, presumably recorded on gas mixtures with constant flow rate and pressure. During the flight, other sources of instabilities can increase the noise of the instrument. However some of the values appear to be lower during the flight (above the planetary boundary layer only). This would need some further explanation.

Conclusions: the advantages and drawbacks of the aerodyne instrument could be better highlighted. The large number of species analysed together is certainly an interesting feature, but it seems to come with increased noise compared to CRDS analysers. Is that really the case or is this a wrong impression coming from an increased in−flight noise which could impact other analysers as well?

Line-by-line comments:

**Page 3**

Line 5: you may clarify that "DLR" in the title is the name of the laboratory owning the spectrometer.

**Page 4**

Line 25: you may keep SLPM for the flow rate, but indicate the value in mL min$^{-1}$ as well

**Page 5**

Line 19: why the use of "cross−calibrated" rather than "calibrated"? Does it involve a particular method?

Line 32: the entire sentence may be rewritten to express more clearly that no dilution was introduced at this stage, which is why you do not need to take into account an uncertainty on the flow rate measurements.

**Page 10**

Line 14−15: what is meant by "not accurately constrained"? There is certainly an issue with the difference in isotopic composition between the sample and the calibration gas, and this aspect deserves a better treatment in the paper. However at this point you are describing the fit of the spectra, and the statement about constraining the isotopic composition of the sample is unclear. Does this mean constraining the fit? The fit window?

**Page 12**

Line 14 "excluding absolute error". Do you mean uncalibrated or expressing the precision only?

Line 15: "values reported by Aerodyne". Which paper? McManus 2011?

Line 2−3: "we were not able to reproduce…" seems a rather negative introduction for a positive result, as everything was made to be insensitive to the cabin pressure. Consider rephrasing.

Comments on figures:

Figure 7 it is not clear if the amount fractions are provided after calibration or not. The legend seems to indicate calibrated values, but the y−axis in the right plot indicates "Methane RAW [ppt]" which would mean raw values before calibration. Please clarify.

Figures 10 and 11: it is too uneasy to compare both analysers on the plots. Differences would be more interesting, as the paper does not include any consideration on the amount fractions of the gases.

Figure 12: $y$−axis of the right plot is the methane amount fraction. Use a symbol and unit such as "$x_{CH4}$ / (nmol mol$^{-1}$)" and indicate in the legend "$x_{CH4}$ is the methane amount fraction".

---

## Author Response (AR2)

**In response to the Associate Editors comments from February the 14th, 2019.**

*Dear Authors, Thank you for your detailed and authoritative responses to the referees' comments. I have carefully reread your well-written paper that certainly merits publication in AMT. At this stage, there are some very minor and technical issues remaining, which should be addressed before publication. Please find these listed below.*

5   Dear Associate Editor,
Thank you very much for your detailed corrections and the careful look over the manuscript. We greatly appreciate your help with this paper. The technical corrections have been implemented in a revised version of the manuscript.

**Minor corrections:**
(page and line numbers refer to the manuscript version that keeps track of revision changes)

10   1. **Some company names appear in capital letters, others do not. It seems preferrable to keep a common format. I suggest capitalizing only the first letter in company names, which is an easier read, especially for long names.**
We have tried to keep everything in a common format now.

**In the same veins, there is excessive use of the brand name to denote the CRDS analysers, but other instruments such as the modified QCLS/ICLs instrument are not presented by insisting on the original brand each time.**
15   **This poses the risk of inadvertent advertising of one particular instrument as compared to others and hinders easy understanding for people outside the field. In general, it is preferable to use brand/model names only when instruments are presented for the first time and when required for the understanding. Otherwise instruments should be referred to by their operational principle or instrument function (eg "hygrometer" instead of "Sensirion", etc.). Please modify Figures 10, 11, 12, 13 and the text accordingly.**
20   We have modified the text and the figures as suggested.

2. **The typesetting guidelines of Copernicus journals require a space between quantity and unit. This also applies to dimensionless units such as the % sign (ie 78 % instead of 78%, ...).**
We hope to have modified the text accordingly at all occurences of values/units.

3. **p 7, l 17-18 : " ... CH4 (Cert.-Nr. CB11361, WMO X2004A for CH4 (Dlugokencky et al., 2005)). C2H6, CO and**
25   **N2O are compared to NOAA ..." -> CO2 is missing in the list even though it is mentioned later on in chapter 5, p 17. Please complete the list.**
$CO_2$ has been added to the list.

4. **p 16, Fig 8 and discussion of frequency lock : There is a regular spike pattern in Fig 8 with a period of about 30/6 5 min. This supports the hypothesis that shifts are not only caused by mere laser drifts, but are also caused by**
30   **some well-timed mechanism. It is not clear whether this is meant by "high-frequency shifts are evident, including discontinuities". The discussion of the spikes could be added to the discussion on p 16.**
We have included the missing reason for these well-timed spikes. These are due to switching over from calibration to sample gas.

5. **p 22, l 20-21 and p 23 Fig 13 : According to the figures, the QCLS-CRDS and the QCLS-FLASK comparison shows**
35   **a good agreement in relative deviations, except for CH4, where QCLS-CRDS gives a 1-sigma of 1.4 ppb, but the flask comparison seems to give a different number (1-sigma = 3.9 ppb (n=40)). This needs to be commented.**
This could be due to different sampling times of the fast QCLS and CRDS observations compared to the slow flasks. We have added this in the main text.

6. **p 21, l 12+, discussion 13CO2 measurement : "We estimate the required isotopic composition that could explain**
40   **a large bias of up to 17 ppm (see Supplement Section 3) in CO2 to be 98.447 % primary isotopologue and 1.079 % secondary isotopologue or $\delta$13C = -19.6 ‰ which seems reasonable according to B. Coplen et al. (2002)." This**

**statement seems to be in direct conflict with the Supplement, where it is derived that d13C must be -37 ‰ to have an impact of 17.2 ppm. Please correct and be more detailed about the -19.6 ‰ and your calibration procedure that should be crucial in infering a particular isotope composition.**

Thank you for pointing us towards this error. The derivation in the Supplement is correct. Here, we wanted to refer to the 10ppm bias found in this study.

**The discussions in the main text and in the supplement also merit clarification. In particular, the definition of x_retrieved remains dubious, as remains the meaning of "possible influence estimate". It might help to differentiate between two effects of opposite sign : HITRAN conventions (a -5 ppm effect when measuring background air CO2) and the systematic bias that could play a role here (up to about +12 ppm). HITRAN assumes 13C isotope abundances of VPDB (d13C = 0) to derive line intensities, and, for the purpose of demonstration, one assumes that these are correct. When we then measure CO2 as 13CO2 we can use the rule of thumb that a relative change of 13C will translate into a relative change of CO2 even if the concentration of CO2 remains unchanged (less 13C leads to less CO2 measured and higher 13C leads to higher values of CO2). This means that the QCL measurement of air with d13C = -9.7 ‰ leads to CO2 that is roughly 1 % (or 4 ppm at 400 ppm CO2) too low (effect of -4 ppm). A positive offset of +10 ppm can only be obtained if the instrument is calibrated with a gas of d13C = -40 ‰. This seems to be reasonable provided the tank CO2 derives from fossil fuel. Air, which has a d13C that is higher than fossil fuel CO2 by 30 ‰ will yield a 0.03 * 400 ppm = +12 ppm higher CO2 abundance. Note also that line 15 on p 3 of the Supplement wrongly refers to methane.**

We tried to change the wording of x_retrieved and "possible influence estimate". To our knowledge HITRAN does not assume 13C isotope abundances of VDPB, instead the natural terrestrial abundances used in HITRAN which are available on the official website are used. Unfortunately we do not quite understand the derivation given above. We tried to give a detailed derivation in the supplement, that we think is complete as is.

**To which degree has the 13C isotopic composition of CO2 to be known to reach WMO compatibility ?**

The requested info has been appended to the supplement.

**Technical corrections:**

(page and line numbers refer to the manuscript version that keeps track of revision changes)

1. **p 1, title** : use subscripts in chemical formulae
   Changed accordingly.

2. **p 1, abstract** : "... and central US." -> "... and central US." (note you use USA without full stops later on ....)
   Changed accordingly.

3. **p 3, l 5** : "since the pre-industrial era, where ..." -> consider replacing "where" by "and"
   Changed accordingly.

4. **p 3, l 11** : "Aircraft provide ..." -> "Aircrafts provide" (eventually "aircraft provides")
   We refer to the plural of Aircraft here. We think it should be Aircraft not Aircrafts.

5. **p 4, l 25** : consider replacing "... required to operate the instrument on research aircraft." by "... required to operate the instrument on a research aircraft."
   Changed accordingly.

6. **p 4, l 27** : "The spectrometer is split into an electronics compartment and an optics compartment." -> "The spectrometer is split into an electronics and an optics compartment.
   Changed accordingly.

7. **p 4, l 28** : "... , etc.." -> "... , etc."
   Changed accordingly.

8. **p 5, l 3** : "... , etc.." -> "... , etc."
Changed accordingly.

9. **p 6, l 2** : "every half of a second." -> "twice per second."
Changed accordingly.

10. **p 6, l 17** : "... avoiding injecting large vibrations into the ..." -> "... reducing vibrations of the ..."
Changed accordingly.

11. **p 6, l 18** : "This translates to a net flow rate of 25 SLPM". Different entities use different standard conditions (IUPAC, NIST, EPA ...) SLPM is therefore not a well defined unit. It would help to recall "your" standard conditions here (just once in the text). Even better, you could (just once) give the molar flow rate in mole/s here. Note that since 1982 IUPAC defines T = 273.15 K and p = 100 kPa as standard conditions.
Changed accordingly.

12. **p 6, l 19** : "when operating with a cell pressure" -> "when operating at a cell pressure"
Changed accordingly.

13. **p 7, l 3** : "Polytetraflouroethylene" -> "polytetraflouroethylene"
Changed accordingly.

14. **p 7, l 16** : "cross-calibrated using a Picarro CRDS against NOAAA standards" -> "cross-calibrated against NOAAA standards using a CRDS "
Changed accordingly.

15. **p 7, l 24** : "Typical calibration distribution diagrams ..." sounds like a technical term. It could be easier to understand using "Histograms of typical calibration measurements are provided ..."
Changed accordingly.

16. **p 8, legend Fig 2** : A letter "H" appears in the drawing, which is not explained in the Fig legend.
Removed clipping of information on this figure

17. **p 10, legend Fig 4** : change all chemical names to small letters, add axis title to x-axis. What is the difference between quantities and units used on the y-axis in this and in the previous (Fig 3) figure ? Shouldn't they have same names and spellings ? Since "adu" does not refer to a proper name, it should be spelled using small letters.
Changed accordingly.

18. **p 11, legends and axis labels Fig 5** : font size is too small
Changed accordingly.

19. **p 11, Fig 5, y-axis label** : The plot does not show the optical depth but the transmittance
Changed accordingly.

20. **p 11, Fig 5, graph labels** : replace "microwindow" by "fit window"
Changed accordingly.

21. **p 11, l 6** : "... shift variables are held constant at the mean over its last 10 values." -> "... shift variables are held constant at their means over the last ten values."
Changed accordingly.

22. **p 13, l 6** : "Eq. (1)" instead of "Eq. 1"
Changed accordingly.

23. **p 13, Table 1** : Broadening coefficient ($\gamma_{air}$)
Changed accordingly.

24. **p 15, l 2** : "Power drawn from the aircraft remained under 50 A at all times and settled at approximately 40 A." Replace "power" by "current" or "electrical current".
Changed to "Electric current ..."

25. **p 15, l 3** : "The volumetric flow rate at ... SLPM". You specify the flow of an absolute amount of gas and not a volume flow. I suggest dropping the term "volumetric".
Removed "volumetric".

26. **p 15, l 9** : "will decrease the signals standard deviation" -> "will decrease the standard deviation of most of the signals"
Changed accordingly.

27. **p 15, Fig 7 left panel** : Legend is difficult to read, please increase font size. Plot could be made more readable by excluding data for $\tau > 500$ s (does not seem significant anyway) and restrict y-scale from 1e-4 to 1e1.
We increased font size of the legend.

28. **p 15, Fig 7 right panel** : It appears that the confidence interval given for linear fit parameters ignores the uncertainties of the x-values. If this is so, it is better not to insist on the confidence intervals of the parameters. Give R-squared with number of digits required to see the deviation from 1.
Changed accordingly.

29. **p 16, Fig 8** : Please increase the thickness of lines in legends. Traces are difficult to identify otherwise.
Changed accordingly.

30. **p 20, l 2** : "(Gordon et al., 2017))" -> "(Gordon et al., 2017)"
Changed accordingly.

31. **p 20, l 15** : put "H2O" in upright letters
Changed accordingly.

32. **p 20, l 18** : Please add + or - sign to bias constants in order to avoid ambiguities, assuming that "+" means that the QCLS is higher than the reference.
Changed accordingly.

33. **p 21, l 22** : "(see 2.3)" -> "(see Section 2.3)"
Changed accordingly.

34. **p 21, l 23** : "Eq. (1)" instead of "Eq. 1"
Changed accordingly.

35. **p 21, l 27-28** : "... precision (uncertainty) estimates based on ambient measurements at stable conditions of 0.3 (2) ppb, 0.02 (0.1) ppm and 2.0 (5) ppb ...". This notation is non-standard and quantities in parentheses might therefore be confused with uncertainties. I suggest writing "... precision/uncertainty estimates based on ambient measurements at stable conditions of 0.3/2.0 ppb, 0.02/0.1 ppm and 2.0/5.0 ppb for ...".
Changed accordingly.

36. **p 23, Fig 13** : y-axis label "Norm occurrence" should come with units in 1/ppb or 1/ppm
Changed accordingly.

37. **p 23, Fig 13, central panel** : symmetrize y-axis (from -2.x to +2.x)
Changed accordingly.

38. **p 23, Fig 13 caption** : Please add legend for colour code of flask samples.
Changed accordingly.

39. **p 23, Fig 13 middle panel on top** : Is there a reason for the skew in the distribution ?
    Changed accordingly.

40. **p 24, l 2** : "(5 to 10mins interval)" -> "(5 to 10 min interval)"
    Changed accordingly.

5 41. **p 24, l 10** : "above the eastern U.S.." -> "above the eastern US."
    Changed accordingly.

[revised manuscript text omitted]

**1 Compatibility & Comparison with other instruments**

We have computed the differences between flasks and the CRDS in the same way they have been computed for the QCLS in the main manuscript by interpolating high frequent in situ data to the flask end fill times. This has been done to get an idea on what spread can be expected due to the long flask sampling time compared to the fast measurement cycles of CRDS and QCLS. Although the spread in Fig. 1 might be slightly bigger for the QCLS compared to the CRDS it remains unclear if this is due to higher sampling rate and/or response time of the QCLS.

[Figure]

**Figure 1.** Comparison of CRDS (upper row) and QCLS (lower row) derived mole fractions to flask samples. Interpretation of the errors against flask samples is difficult for high-variability flight segments, due to the large flask sampling time. The residual plots show color-coded data from 5 typical flights on 10/03/2017, 10/11/2017, 10/14/2017, 10/18/2017 and 10/20/2017.

**2 Cross-sensitivities**

There was doubt if the water vapor correction is deteriorating compatibility between instruments. Figure 2 shows cross sensitivities of the QCLS-CRDS residuals for the flight highlighted in the original manuscript (10/03/2017) with respect to cell pressure, cell temperature and water vapor after correcting for a constant bias. The residuals have been computed by linearly interpolating the higher frequency QCLS to the CRDS time scale, due to different sampling times and patterns. From this figure, we would argue, that the water vapor correction (center row in Figure 2) is not systematically deteriorating compatibility between the instruments.

[Figure]

**Figure 2.** Cross sensitivities of the QCLS-CRDS residuals for the flight highlighted in the original manuscript (10/03/2017) with respect to cell pressure, cell temperature and water vapor after correcting for a constant bias. Due to different sampling times and patterns the higher frequency QCLS data have been linearly interpolated to the CRDS time scale.

**3   Large bias in CO₂**

Explaining the large bias in retrieved $CO_2$ requires an estimation of the influence of the isotopic composition of the working standards  and the sampled air within this study. We have used working standards of synthetic nature from *Air Liquide* due to the large amount of needed calibration gas. Usually these are produced with $CO_2$ from natural gas & oil combustion processes. We determined the $CH_4$ and $CO_2$ values of each working standard gas cylinder using a Picarro G-1301m. This has the drawback that we do not know the isotopic composition of our working standards. The reason why we did not send our working standards to a central lab is because the influence of the isotopic composition had been considered negligible at this stage (Chen et al., 2010). It was only in late summer 2018,  when we found that the instrument was using a $^{13}C^{16}O_2$ line to derive ambient $CO_2$. We assume the large bias originating from differences in isotopic composition in our working standards relative to the natural terrestrial abundances. This requires an estimate on the possible influence which will be given here.

It is commonly assumed that differences in isotopic composition only make up for errors on the order of 0.1 ppm in $CO_2$. This is true if measuring the primary isotopologue, as done with the Picarro CRDS. If the $CO_2$ concentration is derived from the secondary isotopologue ($^{13}C^{16}O_2$) the influence of isotopic composition is much larger. Let us take the $\delta^{13}C = -8.6$

‰ reported for the NOAA standard Cert.-Nr. CB11361 as an example to estimate the influence of isotopic composition on retrieved mole fractions. Per definition $\delta^{13}C$ is given by

$$\delta^{13}C = \left( \frac{R_x}{R_{vpdb}} - 1 \right) \times 1000$$

Inserting the values from above and re-arranging this equation yields

5  $R_{noaa} = \left(1 - 8.6 \times 10^{-3}\right) \times R_{vpdb}$

Inserting the standard ratio for the reference materials of the Vienna Pee Dee Belemnite $R_{vpdb} = 0.011180 \pm 0.000028$ (Tohjima et al., 2009; Chen et al., 2010) yields the corresponding isotopic ratio of

$R_{noaa} = \left(1 - 8.6 \times 10^{-3}\right) \times 0.011180 = 0.011083$

According to HITRAN the primary isotopologue and secondary isotopologue together make up 99.5261 % of atmospheric $CO_2$.
10  To satisfy both equations we obtain an abundance of 0.984350 primary isotopologue and 0.010910 $^{13}C^{16}O_2$ for the NOAA standard dealt with here. However, spectral line intensities $S_{ij}$ as defined on *https://hitran.org/docs/definitions-and-units/* are weighted according to the natural terrestrial abundances reported in HITRAN. Retrieved mole fractions are thus scaled by their terrestrial natural abundance from HITRAN. We'll first compute the unscaled $^{13}C^{16}O_2$ using the natural abundance from HITRAN (98.4204 % primary and 1.1057 % secondary $CO_2$ isotopologue) from a hypothetical 400 ppm background (with a
15  natural isotopic composition as defined in HITRAN) resulting in

$^{13}CH_{4,unscaled}CO_{2,unscaled} = 400 * 0.011057 = 4.4228$

 We can now estimate the influence of the different isotopic composition (NOAA example) from above by scaling the unscaled $^{13}CO_{2,unscaled}$ with the secondary isotopologue abundance computed above. For the given absopriotn line, a retrieval algorithm based on HITRAN will retrieve an abundance of $CO_{2,retrieved}$ according to

20  $x_{retrieved}CO_{2,retrieved} = \dfrac{4.4228}{0.010910} = 405.4$ ppm

From this example we see that the **small perturbation in isotopic composition already has an impact of 5.4 ppm** in retrieved $CO_2$.

Chen et al., 2010 reported on synthetic air with added $CO_2$ from burned petroleum or natural gas with $\delta^{13}C = -37 \pm 11$ ‰. Using this value, and repeating the math from above, we obtain a  **change of 17.2 ppm in retrieved**
25  **mole fractions** resulting solely from a different isotopic composition. Given this estimate, we find that, precise knowledge of the $\delta^{13}C$ of the working standards and the sampled air is needed to enhance $CO_2$ compatibility when operating on the 2227.604 cm$^{-1}$ $^{13}C^{16}O_2$ absorption line. In the abscence of other error sources, achieving WMO compatibility using this absorption line at ambient $CO_2$ concentrations of 400 ppm requires the sum of $\delta^{13}C$ from the working standards and the sampled air to be known better than 4.9 ‰. In reality other error sources are not negligible, which will further reduce the stated margin.

**4 Calibration cycles**

We included distribution diagrams of the calibration cycles during the flight highlighted in the main manuscript (10/03/2017) in Figure 3.

[Figure]

**Figure 3.** Distribution diagrams of the calibration gas measurements during the flight highlighted in the main manuscript on Oct. 3, 2017.

---

## Author Response (AR3)

**In response to Alan Frieds comments from October the 16th, 2018.**

*This is an excellent paper representing very careful and well thought out procedures and analysis methods. The paper is very well written and the results are very sound. I particularly like the fact that two different precision regimes have been identified (within the PBL and above the PBL) and yield different results due to differences in alignment caused by aircraft vibrations.*
5 *We often see this effect in our measurements and highlighting them here is a further illustration of the care devoted to the measurements presented. The one thing that should be added is a brief section indicating how the in-flight precisions were determined. Did the authors base this on the precision of zero air measurements or the precision of ambient measurements under stable conditions? In the case of the latter, ambient variability cannot be ruled out the in-flight precisions may be even better than indicated. I recommend final publication after the following minor points are addressed. As you can see, these are*
10 *all very minor and serve to clarify some of the discussion.*

Dear Alan,
Thank you very much for your kind and helpful comments on the procedures and analysis methods presented herein. The in-flight precisions are in fact based on ambient measurements at stable conditions. Ambient variability can thus not be completely ruled out, however in-flight precision figures could only benefit from sampling gas from pressure cylinders. We further
15 completely agree, that a pressure-stabilized enclosure would be the ultimate solution. This would imply heavy modifications on the instrument and render our acquired FAA certification (required for all European Research Aircraft) invalid. For this reason it has not yet been done at this point.

1. **Introduction, Line 6: change the word "remain" to "have"**
   The wording has been changed to:
20   "[...] is a strong greenhouse gas and is expected to have the most important ozone-depleting anthropogenic impact throughout [...]"

2. **Introduction, Line 16: Since there have been extensive measurements of atmospheric gases well before QCLs and ICLs in the mid-IR using for example, liquid nitrogen cooled lead-salt diode lasers as well as other sources, a brief sentence giving a reference to some of this work should be included. One can cite numerous sources, but**
25   **one convenient way (at the risk of being self-serving) would be to cite our text book chapter which has many of these references (Chapter 2: Infrared Absorption Spectroscopy by A. Fried and D. Richter, in the book Analytical Techniques for Atmospheric Measurement, edited by D.E. Heard, Blackwell Publishing, 2006).**
   A brief sentence giving a reference to some of this work has been added:
   "[...] Spectroscopic instruments making use of molecular ro-vibrational absorption allow for high temporal coverage
30   through fast instrument response times (Chen et al., 2010). Some have already been used for airborne research, e.g. established IR spectrometers (O'Shea et al. (2013); Santoni et al. (2014); Cambaliza MOL (2015); Filges et al. (2015)). Significant effort led to instruments operating in the mid infrared (IR) region, e.g. liquid nitrogen cooled lead-salt diode laser based spectrometers (Fried and Richter, 2007). With the commercial availability of continuous-wave lasers emitting in the mid IR region near ambient temperature (Capasso (2010); Vurgaftman et al. (2015); Kim et al. (2015), Beck et al.
35   (2002)) several new instrument designs have emerged (McManus et al. (2015); Zellweger et al. (2016)). [...]"

3. **Introduction, Line 20: Please change the wording "custom-built QCL..." to "custom-built difference frequency generation (DFG) absorption spectrometer"**
   The relevant sentence has been changed to:
   "[...] reported on a custom-built difference frequency generation (DFG) absorption spectrometer [...]"

40 4. **Introduction, Line 27: Are you strictly referring to established cavity ring-down instruments here or are you referring to more generally IR absorption instruments? Since you mention the "described spectrometer", I think you should change "established cavity ring down" to "established IR spectrometers"**
   We are referring to established IR spectrometers in general. The wording has been changed accordingly to:
   "[...] used for airborne research, e.g. established IR spectrometers [...]"

5. **Page 4, Line 25: It would be useful to the reader to further elaborate on the meaning of "jeopardized nominal system startup". Do you simply mean the large in rush current to get the pump going is more than the airplane circuit breakers can take, or does this mean that this may cause damage to other parts of the instrument?**

   Inrush currents have previously been too large, causing circuit breakers to trigger. Depending on whether computer/data analysis systems sharing the same circuit breaker, were already turned on, sudden power loss has previously implied further consequences. However, the few power loss situations we have experienced with this instrument have had no noticeable effect. The following sentence has been added for clarification:

   "[...] nominal system startup (priv. comm. Stefan Müller, MPI Mainz). Sudden power failure, due to over-current triggering aircraft circuit breakers, may lead to failures in the data analysis equipment. [...]"

6. **Figure 5: It would be very helpful to the reader to indicate the mixing ratios in the figure caption used in recording the various spectra.**

   The mixing ratios will be indicated for each species in the three microwindows in a revised version of this manuscript.

7. **Page 10, Line 2: Where is the weak CH4 line relative to the C2H6 line which is used for spectral shifting of the C2H6 feature?**

   In a revised version of this manuscript line 1/2 on Pg. 10 has been changed to: "A single adjacent $CH_4$ line, located at $2989.981\,cm^{-1}$ has been included in order to obtain good $C_2H_6$ data even under these challenging conditions."

8. **Page 11: This is a very nice discussion of the various broadening parameters and how they are handled. However, this reviewer wonders how important actually including the self-broadening and water broadening are in the final fits since these are smaller by the fact the sampling pressure is 50-mb and the overall spectral stability is in the $10^{-3}cm^{-1}$ range? The air broadening at this pressure is only $\sim 0.0035cm^{-1}$ which is close to the spectral stability. Perhaps a brief mention of how the inclusion of self and water broadening changes the retrieved results should be included.**

   Not including the self and water foreign broadening leads to relative errors in the range of 0-2%, depending on the species of interest. While small for $C_2H_6$ and $CH_4$ with $< 0.03\%$, the influence on retrieved CO is rather large with $\sim 2\%$.

   "[...] Not including the self and water foreign broadening leads to relative errors in the range of 0-2% for the described setup, depending on the species of interest. While small for $C_2H_6$ and $CH_4$ with $< 0.03\%$, the influence on retrieved CO is rather large with $\sim 2\%$. [...]"

9. **Figure 7 and Its Caption along with Page 13, Line 1: At the FLAIR (Field Laser Applications in Industry and Research) the Program Committee strongly recommended that references to "Allan Variances" should be denoted "Allan-Werle Variances" in honor of the late Peter Werle who adapted this concept to atmospheric measurements. Below I include a portion of the Program Committee's Obituary for Peter Werle and its recommendation (this need not be included in the final paper but is included here for your reference). Also, what mixing ratios were used in recording Fig. 7 (zero air or calibrated standard mixing ratios)?**

   Thank you very much for pointing this out. Occurrences of "Allan variance" have been changed accordingly in a revised version of this manuscript.

10. **Page 15, Discussion of Fig. 10: In comparing flask and in situ measurements it should be mentioned that care must be exercised in that during times of rapidly changing ambient mixing ratios one may not get agreement between the slow flask samples and fast in situ measurements. Although this is obvious, it is worth mentioning here. I see this is discussed in the Fig. caption 13 but it is also worth mentioning here.**

    The following short sentence has been added to the manuscript:

    "[...] both sampled through an upstream dryer. It should be noted, that care must be taken when interpreting the differences between slow flask samples and fast in situ measurements for high-variability flight segments. [...]"

11. **Page 18, in the discussions of cabin pressure dependence: The authors should mention that it is not possible to accurately compare the dependence of one instrument relative to another since many instrument-dependent**

**and other factors come into play. For example, some of the dependence is due to the changing mixing ratios for the species under study in the open-air path. Additional dependencies result from movement of optical windows and other components and are instrument dependent. Also, we find that the rate of cabin pressure change is an important factor, and this is specific to the particular aircraft and the flight pattern employed. Hence, the left side of Fig. 12 may not tell the whole story. We find that the delta Pcabin/delta time comes into play between zero acquisitions, and I would expect the same thing here. Perhaps a comment on this should be mentioned.**

The relevant text portion has been rephrased to:

"[...] A severe cabin pressure dependence in excess of $0.3\,ppb\,hPa^{-1}$ in $CH_4$ mixing ratio has been previously reported for airborne TILDAS instrumentation (Pitt et al., 2016). This instrumentation however physically differs from the one reported here. It is not possible to accurately compare the dependencies of one instrument relative to another since many factors/quantities involved are instrument-specific, e.g. the open-path length, the positioning and properties of optical elements, like windows and mirrors, the stiffness and thermal expansion coefficients of employed optical stands, etc.. We were nevertheless able to effectively minimize cabin pressure dependencies during operation of the QCLS instrument aboard the C130 using the calibration strategy from Sect. 2.3. [...]"

**In response to Anonymous Referee #2 comments from November the 9th, 2018.**

*This paper describes the performances of an analyser of the major greenhouse gases in air on board of an aircraft. Details are provided on the analyser hardware, the analytical software and the calibration method, followed by an evaluation of the performances by comparison with other analysers present during the same flight. While the analyser itself is not new and was already described in a previous paper (McManus 2011), in this work the number of analysed compounds was extended, the in situ performances were looked at more deeply, and the calibration method was improved. The paper is generally well written, well-structured, clear, and provides lots of details on the instruments and methods. However the section on performance evaluation needs some more work, both in its content and format. I therefore recommend a minor revision before the paper can be published in AMT.*

*Comments on the terminology*
*- Units to be written in plain (not italic) format - The format to display a value with its unit is "value-space-unit" for example "204m" on page 4 should be written "204 m"*
*- "mixing ratio" to be replaced by "amount fraction", expressed in mol mol-1 (nmol mol-1 for ppb, μmol mol-1 for ppm).*
*- Names of molecules to be written in plain (not italic) format.*
*- Allan deviation seems to be confused with Allan variance. When values are reported in the same unit as the concentrations, this should be a deviation. Please check the correct usage over the document*

Dear Referee,
Thank you very much for the detailed and very helpful comments and for the time spent on reading and reviewing this manuscript. We greatly appreciate it. The comments made on terminology were of great help and have been implemented in a revised version of the manuscript. We will directly follow up on the specific comments.

**Specific comments by section:**

1. **Section 2.1: the text describes two sealed cells containing CH4 and N2O. Where are they on Figure 1? Please indicate the purity of the gas and its pressure.**
There is only a single sealed cell containing $CH_4$ and $N_2O$. Its position has been marked in a revised version of Fig. 1. The gas inside the cell has an approximate pressure of $3500\,Pa$. The gas does not need to be pure. As the laser scans over the absorption features of $CH_4$ and $N_2O$ the laser can be spectrally referenced to the relevant molecular absorption lines, which is the single purpose of the sealed cell.

2. **Section 2.3: please provide more information on the calibration mixtures. In particular NOAA standards are all identified within NOAA database and you could just provide their reference to allow the users looking at all values measured by NOAA. At least please indicate the nominal amount fractions, their uncertainties, and the isotopic composition for CO2. This last value is of importance as you noticed a bias between the CO2 amount fractions measured with your instrument and those measured with the PICARRO.**
We included the requested details on the used NOAA standards for CH4 and CO2. However, we have to note, that we used working standards of synthetic nature from *Air Liquide* due to the large amount of needed calibration gas. Usually these are produced with $CO_2$ from natural gas & oil combustion processes. We determined the $CH_4$ and $CO_2$ values of each working standard gas cylinder using a Picarro G-1301m. This has the drawback that we do not know the isotopic composition of our working standards. The reason why we did not send our working standards to a central lab is because the influence of the isotopic composition had been considered negligible at this stage (Chen et al., 2010). It was only in late summer 2018, that we found out (during development of JFIT) that the instrument was using a $^{13}C^{16}O_2$ line to derive ambient $CO_2$. We assume the large bias originating from differences in isotopic composition in our working standards relative to the natural terrestrial abundances.
"[...] The cylinders have been cross-calibrated against NOAA standards and are thus traceable to World Meteorological Organization (WMO) standards for $CH_4$ (Cert.-Nr. CB11361, WMO X2004A for $CH_4$ (Dlugokencky et al., 2005)) [...]"

"[...] In this study we used working standards of synthetic nature from *Air Liquide*. Usually these are produced with $CO_2$ from natural gas & oil combustion processes. We determined the $CH_4$ and $CO_2$ values of each working standard gas cylinder using a NOAA-anchored (Cert.-Nr. CB11361) Picarro G-1301m. This has the drawback that we do not know the isotopic composition of our working standards as its impact had been considered negligible, e.g. (Chen et al., 2010). We only found out during development of JFIT, that the instrument is using a $^{13}C^{16}O_2$ line to derive ambient $CO_2$. We estimate the required isotopic composition of such a $CO_2$ to be 98.447% primary isotopologue and 1.079% secondary isotopologue or $\delta^{13}C = -19.6$ ‰ which seems reasonable according to B. Coplen et al. (2002). Since we are reporting retrieved mole fractions relative to the WMO scale, only the working standard reproducibility contributes to the total uncertainty of $CH_4$. Uncertainty on $CO_2$ is difficult to assess here because of the unknown isotopic composition in our working standards. [...]"

3. **Section 4, ground-based performance: the reported Allan deviations seem a bit large. Compared to McManus 2011 on CO 2 for example, a factor 10 is noted. Please consider revising the statement that "values are in good agreement with the values reported by Aerodyne" and/or provides further support. Is there an effect of the calibration system described in 2.3, which is said to be used to check the stability and the linearity?**

The lasers frequency reported in McManus 2011 differs from the emitted frequency reported in this publication. Therefore it is not possible to directly compare with the values reported in McManus 2011. Here we are referring to the specifications reported by Aerodyne Inc. for this particular instrument. Nevertheless we removed this sentence as it is not necessary at this point. Although the calibration system has a strong influence on the instruments accuracy, we see no variation in precision after carefully looking at signal changes before and after calibration versus similar intervals between calibrations.

4. **Section 5: while the traceability of measurements with the QCL is clear (calibration with NOAA standards), nothing is indicated regarding the PICARRO. This is needed to fully understand the origin of biases. It seems that an anchored to NOAA is assumed, but this deserves further details (which standards? How many calibration steps? Isotopic composition?). When both instruments are compared, it would be more useful to express the difference in amount fraction, both in the text and in the graphs. This should then be compared with their uncertainties, not taking into account common sources of uncertainties such as NOAA uncertainty if all amount fractions are expressed on the same scale. Going beyond this, some consideration on how this compares with the Data Quality Objectives set by WMO would be of interest. The treatment of the constant bias found between the AERODYNE and the PICARRO analysers needs to be improved. Are both analysers calibrated directly with NOAA standards? How different are the calibration gases? It would be valuable to estimate the bias one could expect from the isotopic difference, as done for example in the paper of Chen et al. (Atmos. Meas. Tech., 3, 375–386, 2010), and compare with the observed bias. Indeed, an observed bias of 10 μmol mol -1 seems very large.**

The PICARRO instrument is anchored to NOAA. CO2 is WMO X2007, CH4 is WMO X2004A, CO is WMO X2014A. It is calibrated hourly during flights using a fixed standard and weekly using a three-point calibration with high, low and target calibration standards. The corresponding references have been implemented in a revised version of the manuscript. Concerning the second part of this comment: it is important to know the dynamic range that is covered when looking at differences between instruments. Vanishing differences at vanishing dynamic range do not tell the whole story about instrument performance. We included both, the differences (as histograms in Fig. 13) and the absolute values (dynamic range) in Fig. 10 and 11. The origin of the biases is not yet fully understood. It was suggested that water vapor correction could have an impact on this. The reason for this assumption is that the calibration standards are always dry, whereas sampled air is not dried before entering the sample cell. Correlation plots however show no signfcant influence of water vapor on the residuals between the dry-air-sampling Picarro and the QCLS. It is therefore very unlikely that the water vapor correction is the source of the large bias in $CO_2$. Instead we identified the difference in isotopic composition of the calibration standard versus sampled atmospheric air as the most probable cause. Chen et al. (Atmos. Meas. Tech., 3, 375–386, 2010) estimated the influence for a Picarro greenhouse gas analyzer measuring the primary $CO_2$ isotopologue. It is commonly assumed that the influence of isotopic composition is on the order of 0.1 ppm. Using the $^{13}C^{16}O_2$ line at $2227.604 \, \text{cm}^{-1}$ via HITRAN-based direct absorption spectroscopy, we estimate a much larger influence, that could well explain the bias encountered (see above). We estimated the required isotopic composition of such a $CO_2$ to be 98.447%

primary isotopologue and 1.079% secondary isotopologue ($^{13}C^{16}O_2$). We included this in the text. It is therefore one of the major findings of this study, that knowledge on isotopic composition of the calibration standards is of paramount importance when using the mentioned absorption line.

"[...] In situ $CH_4$, $CO_2$, and $CO$ were measured using a PICARRO G2401-m cavity ring-down spectrometer, and in situ $CO_2$, $CH_4$, and $H_2O(g)$ were measured using a PICARRO G2301-m cavity ring-down analyzer. Both PICARRO instruments are anchored to WMO X2007 for $CO_2$ (Zhao and Tans, 2006), WMO X2004A for $CH_4$ (Dlugokencky et al., 2005) and WMO X2014A for CO (Baer et al., 2002). [...]"

"[...] In this study we used working standards of synthetic nature from *Air Liquide*. Usually these are produced with $CO_2$ from natural gas & oil combustion processes. We determined the $CH_4$ and $CO_2$ values of each working standard gas cylinder using a NOAA-anchored (Cert.-Nr. CB11361) Picarro G-1301m. This has the drawback that we do not know the isotopic composition of our working standards as its impact had been considered negligible, e.g. (Chen et al., 2010). We only found out during development of JFIT, that the instrument is using a $^{13}C^{16}O_2$ line to derive ambient $CO_2$. We estimate the required isotopic composition of such a $CO_2$ to be 98.447% primary isotopologue and 1.079% secondary isotopologue or $\delta^{13}C = -19.6\,‰$ which seems reasonable according to B. Coplen et al. (2002). Since we are reporting retrieved mole fractions relative to the WMO scale, only the working standard reproducibility contributes to the total uncertainty of $CH_4$. Uncertainty on $CO_2$ is difficult to assess here because of the unknown isotopic composition in our working standards. [...]"

5. **Section 5, uncertainties: it is not so common to see combined uncertainties considered in such measurements, and the effort of the authors is certainly valuable. However some consideration on how these values compare with other instruments would be required. Is the calibration procedure specific to this instrument? Does this imply a larger uncertainty than for others? Would you say this instrument has comparable precisions than others?**

The calibration procedure described herein is not instrument specific. It could be applied to other in situ instruments as well. Regarding the accuracy involved, there is always a trade-off between measurement time and accuracy: Increasing the number of calibration cycles improves achievable accuracy at the cost of observation time. It does not imply a larger uncertainty than others, as we do not use the online calibration mixing with the MFCs when taking data (Online mixing would add the uncertainty on the mass flow controllers on top). The instrument described herein is unique in that it offers many simultaneously observed species. It may be possible to find instruments showing better precision figures measuring a single or two species but we seriously doubt, that any other instrument with those many species sampled simultaneously will show better precision figures. Furthermore, as described in the text we estimate the uncertainty on calibration sequence evaluation with $2\sigma$, which is again a worst-case assumption. Unfortunately we had a numerical error in the first version of the manuscript and the values listed in Tab. 3 were not double the precision. We corrected this in the revised version of the manuscript. We further included a short sentence on precision comparison with the available PICARRO instrument:

"[...] Precision (uncertainty) figures given in Tab. 3 can be compared to 2s-1$\sigma$ PICARRO G2401-m airborne precision (uncertainty) estimates based on ambient measurements at stable conditions of 0.3 (2) ppb, 0.02 (0.1) ppm and 2.0 (5) ppb for $CH_4$, $CO_2$ and CO, respectively. [...]"

6. **Section 5, discussion on instruments precisions: Allan deviations (not variance) were measured before the flight and during the flight. It is not very clear how those values compare. One would expect the lowest values during ground-based measurements, presumably recorded on gas mixtures with constant flow rate and pressure. During the flight, other sources of instabilities can increase the noise of the instrument. However some of the values appear to be lower during the flight (above the planetary boundary layer only). This would need some further explanation.**

Allan variances were not measured during flights. The in-flight precision values are instead based on ambient measurements at stable conditions. Ambient variability can thus not be completely ruled out. Meaning, we are looking at the worst case scenario here. Thank you for pointing us towards this mistake on the ground-based precision values. The values reported for ground based operation were based on an older version of the retrieval software. We have corrected this in a revised version of the manuscript.

"[...] Typical in-flight precision figures based on ambient measurements at stable conditions for both regimes [...]"

7. **Conclusions: the advantages and drawbacks of the aerodyne instrument could be better highlighted. The large number of species analysed together is certainly an interesting feature, but it seems to come with increased noise compared to CRDS analysers. Is that really the case or is this a wrong impression coming from an increased in-flight noise which could impact other analysers as well?**

Here, we do not want to compare the two instruments against each other. Instead our goal is to demonstrate the suitability of the described instrument, given the calibration approach and post-processing described herein, for airborne observation with the ultimate goal of inferring local to regional fluxes. The instrument has advantages and drawbacks when directly compared to CRDS analyzers. One of the drawbacks is the reduced absorption path length and the resulting lower precision, aswell as the large amount of calibration gas necessary for 10 % of the measurement time. A big advantage is the simultaneous measurement of all targeted species. There is practically no dead time in between measurements, which is especially useful in close vicinity to sources and/or for young weakly dispersed (spatially narrow) plumes. This instrument sees everything, while there is a certain chance with sequentially probing instruments of missing a narrow plume or not getting the peaks right. This is further described in Sect. 5. The large number of observed species is another big advantage that can be used for source attribution.

**Line-by-line comments:**

1. **Page 3, Line 5: you may clarify that "DLR" in the title is the name of the laboratory owning the spectrometer.**
   The relevant sentence has been changed to include a definition of DLR:
   "[...] The spectrometer system used here builds upon the *Dual Laser Trace Gas Monitor*, a commercial tunable IR laser diode absorption spectrometer (TILDAS) available from *AERODYNE RESEARCH INC., Billerica, USA*, acquired by *Deutsches Zentrum für Luft- und Raumfahrt* (DLR) in late 2016. [...]"

2. **Page 4, Line 25: you may keep SLPM for the flow rate, but indicate the value in mL min-1 as well**
   We don't really see the benefit of reporting flow rates in mL min-1, but 23 SLPM would yield 23000 mL/min at standard conditions ($p = 101325\,\mathrm{Pa}$, $T = 273.15\,\mathrm{K}$). We thought about converting SLPM to SI units $1\,\mathrm{SLPM} = 1.68875\frac{\mathrm{Pa\,m^3}}{\mathrm{s}}$ but we omitted this, because we assumed SLPM to be a commonly used unit for in situ measurements.

3. **Page 5, Line 19: why the use of "cross-calibrated" rather than "calibrated"? Does it involve a particular method?**
   Here, we want to express the fact, that we calibrate our working standards using a Picarro G1301-m to NOAA standards as described above. This is what we refer to with "cross-calibration".

4. **Page 5, Line 32: the entire sentence may be rewritten to express more clearly that no dilution was introduced at this stage, which is why you do not need to take into account an uncertainty on the flow rate measurements.**
   The relevant text portion has been rephrased to:
   "[...] The online mixing feature is not used for in-flight calibration. Hence, no dilution of the calibration standard with zero air is introduced during flights and the uncertainty on the flow rate measurements can be omitted. Online mixing (relevant for linearity checks) adds the uncertainty of the controlled mass flow on top of the gas cylinder uncertainties. [...]"

5. **Page 10, Line 14-15: what is meant by "not accurately constrained"? There is certainly an issue with the difference in isotopic composition between the sample and the calibration gas, and this aspect deserves a better treatment in the paper. However at this point you are describing the fit of the spectra, and the statement about constraining the isotopic composition of the sample is unclear. Does this mean constraining the fit? The fit window?**
   We agree that this information is not needed at this point for describing the spectral fit. It is dealt with in Section 5. We thus removed the complete sentence.

6. **Page 12, Line 14 "excluding absolute error". Do you mean uncalibrated or expressing the precision only? Line 15: "values reported by Aerodyne". Which paper? McManus 2011?**

Here, we state the 1-sigma precision and measurement frequency only. The absolute error stated by the pressure trans-ducers manufacturer is stated with $\pm 0.5\,hPa$. The relevant text portion has been rephrased to:

"[...] The volumetric flow rate stabilized at $23\,SLPM$ for a sample cell pressure regulated at $50.0\pm 0.5\,hPa$ ($0.2\,hPa$ precision @ $5\,Hz$). [...]"

7. **Page 18, Line 2-3: "we were not able to reproduce…" seems a rather negative introduction for a positive result, as everything was made to be insensitive to the cabin pressure. Consider rephrasing.**
The relevant text portion has been rephrased to:

"[...] A severe cabin pressure dependence in excess of $0.3\,ppb\,hPa^{-1}$ in $CH_4$ mixing ratio has been previously reported for airborne TILDAS instrumentation (Pitt et al., 2016). This instrumentation however physically differs from the one reported here. It is not possible to accurately compare the dependencies of one instrument relative to another since many factors/quantities involved are instrument-specific, e.g. the open-path length, the positioning and properties of optical elements, like windows and mirrors, the stiffness and thermal expansion coefficients of employed optical stands, etc.. We were nevertheless able to effectively minimize cabin pressure dependencies during operation of the QCLS instrument aboard the C130 using the calibration strategy from Sect. 2.3. [...]"

**Comments on figures:**

1. **Figure 7 it is not clear if the amount fractions are provided after calibration or not. The legend seems to indicate calibrated values, but the y-axis in the right plot indicates "Methane RAW [ppt]" which would mean raw values before calibration. Please clarify.**
The depicted methane amount fractions are indeed raw signals before calibration. A synthetic calibration gas has been mixed from zero and calibration gases using the described calibration system, in order to verify the linearity of retrieved amount fractions.

2. **Figures 10 and 11: it is too uneasy to compare both analysers on the plots. Differences would be more interesting, as the paper does not include any consideration on the amount fractions of the gases.**
It is important to know the dynamic range that is covered when looking at differences between instruments. Vanishing differences at vanishing dynamic range do not tell the whole story about instrument performance. We included both, the differences (as histograms in Fig. 13) and the absolute values (dynamic range) in Fig. 10 and 11.

3. **Figure 12: y-axis of the right plot is the methane amount fraction. Use a symbol and unit such as "xCH4 / (nmol mol-1)" and indicate in the legend "xCH4 is the methane amount fraction".**
xCh4 is commonly used for total column measurements. We therefore refrain from changing the axis label here.

**In response to Anonymous Referee #4 comments from October the 30th, 2018.**

*The paper by Kostinek et al. presents ground-based and in-flight performance assessments of a commercial trace gas analyzer (TILDAS, Aerodyne Research Inc.) after its adaptation for airborne operation. The subject is highly topical and targets a key issue that every scientist is facing when taking decision on analyzer selection. This can even be critical considering the*
5   *stringent place and measurement-time limitations of flight campaigns. Here, the author's choice is on a dual-laser direct absorption spectrometer with multi-species (i.e., five compounds) detection capabilities deploying state-of-the-art mid-IR laser sources (both QCL and ICL). The in-flight intercomparison with CRDS-based instruments and flask samples is an important element of the manuscript that can be of interest for the community involved in airborne measurements. The manuscript is well written, and I recommend publication after addressing some comments and changes listed below.*

10   Dear Referee,
Thank you very much for your careful review and the detailed, helpful comments. We greatly appreciate all your work involved with reviewing this manuscript. The comments include very interesting thoughts and insights. Especially those on instrument details sparked some new ideas to further improve the instrument performance in the future. The technical level of the specific comments is quite high. We hope to have answered to your full expectation.

15   **General comments:**

1. **The title should better reflect the content of the manuscript. Given that for the measurements the spectrometer is equally using both QCL and ICL devices, this should be weighted the same. Furthermore, the instrument was mainly adopted and not modified for airborne operation. Thus, my suggestion is to write: "Adaptation and performance assessment of a dual-laser mid-IR direct absorption spectrometer for ..."**

20   We agree, that the title should weigh QCL and ICL the same, we thus modified the title to include both after the "initial decision" phase. Based on your suggestion from above we further changed the title to "Adaptation and performance assessment of a Quantum / Interband Cascade Laser Spectrometer for simultaneous airborne in situ observation of CH4, C2H6, CO2, CO and N2O"

2. **The abstract should focus on summarizing briefly the highlights and findings of the presented work. Thus, I sug-**
25   **gest starting directly with L7, "Here we demonstrate..."**
We modified the abstract based on this comment:
"Tunable laser direct absorption spectroscopy is a widely used technique for in situ sensing of atmospheric composition. Aircraft deployment however, poses a challenging operating environment for instruments sensing climatologically-relevant gases in the Earth's atmosphere. Here, we demonstrate the successful adaption of a [...]"

30   3. **Although the introduction contains a brief hint about the large number of available measurement techniques for airborne atmospheric measurements, it is unfortunate that the authors completely refrain to motivate their choice for a particular analyzer. Clearly there are some benefits of having multi-species capabilities at slightly higher sampling rate, but how this compensates for the obvious limitations, such as cabin pressure dependence, frequent high-flow calibration requirements, tedious post-processing of the raw spectral data, high power and calibration**
35   **gas consumption, etc.? A more elaborate discussion on advantages/disadvantages of the chosen approach would significantly improve the manuscript.**
We have added a brief statement in the introduction on the motivation for the particular analyzer used. As stated in the response to reviewer #2: "We consider it to be out of the scope of this paper to directly compare the two instruments against each other. Instead our goal is to demonstrate the suitability of the described instrument, given the calibration
40   approach and post-processing described herein, for airborne observations with the ultimate goal of inferring local to regional fluxes. The instrument has advantages and drawbacks when directly compared to CRDS analyzers. One of the drawbacks is the reduced absorption path length. A big advantage is the large number of simultaneously sampled species using a single instrument. Depending on the scientific objective of the aircraft campaign, the additional measurements can either facilitate source attribution of observed methane enhancements (e.g. using ethane and/or nitrous oxide)), or

will allow to study specific scientific questions related to N2O, since there is no dead time in between each single measurement. This is especially useful for measurements in close vicinity to sources where plumes are only weakly dispersed (spatially narrow). While there is a certain chance of missing narrow plume structures with sequentially probing instruments, this QCL/ICL spectrometer does not suffer from such a problem. This is further described in Sect. 5. We included the motivation for the choice:

"[...] This particular analyzer has been favored over other instruments for its simultaneous multi-species capability and its sampling pattern, allowing the detection and quantitative observation of spatially narrow plumes. [...]"

4. **In the same context, it is also not obvious in the present form, why the authors decided for an extensive calibration scheme and additional data post-processing instead of developing a purged and sealed enclosure around the instrument. Apparently, most of the relevant drifts or biases are due to ambient air (H2O mainly) absorptions outside the multi-pass cell.**

We do agree that a pressure-stabilized and zero-air-flushed compartment would certainly be a significant improvement. However, this might be a simpler task for ground- or lab-based instruments but it is far more challenging for airborne instrumentation. With the current setup we are at the absolute mass limit to be able to acquire FAA certification. An additional pressurized compartment would at least mean some additional weight, not to mention the required flushing gas. It is also unsure if such a compartment would solve most of the problems. We fear that this might improve things but not completely solve all related problems because of the very sensitive apparatus involved and the harsh environment aircraft deployment poses. In this context we agree with Alan Fried, who commented on this in a previous review: "For example, some of the dependence is due to the changing mixing ratios for the species under study in the open-air path. Additional dependencies result from movement of optical windows and other components and are instrument dependent. Also, we find that the rate of cabin pressure change is an important factor, and this is specific to the particular aircraft and the flight pattern employed."

5. **Considering that the spectral retrieval software of the manufacturer has been around for more than 20 years, supporting a large number of custom-built instruments applied in a wide range of applications, one would assume that the software experienced a continuous development and incorporates many fine-tuning/customizing features in order to optimize also the fitting process. Therefore, it is highly interesting and valuable if the custom retrieval software (JFIT) significantly improves the performance. This must, however, be more clearly documented by a side-by-side comparison of the results of both software packages. Especially, the additional shift parameters introduced by JFIT and its co-allocation to various segments of the spectral window seems rather subjective and should be quantified in terms of performance improvements. The authors refer to Fig.8 (p13) to show that the tuning rate of the lasers is stable, which seems to be in contradiction with the many shift parameters introduced in JFIT.**

The spectral retrieval software provided by the manufacturer certainly is a very powerful tool and as you say, it implements lots of tuning and tweaking features that can be used for optimization of the retrieval process. We do not claim to provide a better software package, instead we purposely wrote our own retrieval software mainly to learn about possible modifications and to be able to easily adapt our code to new problems. The handling of the shift parameters is one such example. We are not sure of how that could have been done other than writing our own software. We think that it makes perfect sense to split the shift parameters on the spectral axes. Even though spectral shifts seem perfectly co-linear tiny shifts can have significant influence on the retrieved mixing ratios.

6. **Similarly, the discussion of the water vapor correction should contain further details about the observed biases and drifts in the whole range of water concentrations experienced during flights. Measuring water concentration at absolute level is challenging, so the authors should show the observed correlation between generated humidity and spectroscopically measured water mixing ratios. Also, some discussion is required to make clear how the additionally introduced broadening effect improves the measurements, and compare this to the impact of the significant spectral instability (10-3 cm-1) and potential temperature fluctuations of the sampled gas and of the cabin during flights.**

We are aware that measuring water concentrations at absolute levels certainly is a difficult task. We purposely do not

mention water vapor in the title because for us water vapor is just a side product used to enable reporting dry-air mixing ratios. The referred section mainly describes how the water vapor foreign broadening coefficient is obtained. This does not involve measuring water vapor at absolute levels, instead it is only necessary to span the range of atmospheric $H_2O$. The correlation for actual atmospheric measurements can be seen from the lowest panel in Fig. 10. We slightly modified the section to make this more clear and also to discuss how the broadening effect improves the measurements.

"[...] partial pressure of water vapor $p_{H_2O}$ and the water broadening coefficient $\gamma_{H_2O}$. The former can be computed from the measured water vapor concentration. The latter can be empirically determined. Not including the self and water foreign broadening leads to relative errors in the range of 0-2% for the described setup, depending on the species of interest. While small for $C_2H_6$ and $CH_4$ with $< 0.03\%$, the influence on retrieved CO is rather large with $\sim 2\%$. In order to obtain $\gamma_{H_2O}$ two MFCs are used to modify mixing ratios of water vapor in a clean and dry calibration gas. This does not involve measuring water vapor at absolute levels, instead it is only necessary to span the range of atmospheric $H_2O$. An additional downstream pump allows, in combination with a manually-controlled needle-valve, tuning the absolute pressure at the instrument inlet to simulate altitude changes. For these tests, the QCLS instrument has been operated at low flow rates of approx. $1\,SLPM$ due to limitations on the two mass flow controllers. The water broadening coefficient $\gamma_{H_2O}$ has been adjusted iteratively until reported dry-air mixing-ratios of the species of interest remained constant for the set of water vapor mixing ratios. [...]"

7. **Since the instrument is a unique platform, where two different mid-IR laser sources are operated side-by-side, a more detailed comparison of performances and noise characteristics of QCL/ICL would certainly be an added value to the manuscript.**

We sincerely agree, that this would be a very interesting analysis and a nice added value to the manuscript. However, this kind of study is out of the scope of this paper. This is something that should be carried out by laser experts and not as an addendum to this paper. We think this to deserve a publication on its own.

**Specific comments:**

1. **Pg2, L27: need more clarification what is meant by sequential and truly concurrent sensing. Otherwise, there should be a short note about the importance/benefit of measuring at 0.5 Hz instead of 2 Hz.**

In this context "concurrent sensing" is used to describe the fact, that the individual species are measured quasi-simultaneaously. The laser sweeps over the absorption lines with a frequency of 1.5kHz, resulting in a sampling of the spectral absorption of all targeted species at the same frequency. This is to be seen as opposed to "sequential" measurements, where one species is sampled after the other. Further details on the "sequential" approach are given in Section 5. To clarify this, we changed the relevant sentence to:

"[...] Unlike many established instruments measuring different species sequentially (one species after the other), the described spectrometer allows for concurrent sensing of the selected species and faster response times. Fast system response times are valuable to resolve the high variability in trace gases near strong sources. [...]"

2. **Pg2, L25: a reference to the paper at this stage is enough; especially, that Section 2 starts with the same information.**

The relevant sentence was redundant and has been removed in a revised version of the mansucript.

3. **Pg2, L29-31: How to interpret these cited works? In the present context, they give the impression that there is no open question regarding the suitability of QCLS for airborne measurements.**

We rephrased the corresponding text in the introduction to:

"[...] . Santoni et al. (2014) describe the successful deployment and evaluation of a similar airborne spectrometer (Harvard QCLS) for more than 500 flight hours. In contrast, Pitt et al. (2016) reported a severe cabin pressure dependency of their $N_2O$ and $CH_4$ measurements using a commercial instrument (Aerodyne QCLS). By implementing a pressure-differentiated calibration method they were able to correct the corresponding data set, but had to omit roughly half of the measured data. Recently, Gvakharia et al. (2018) reported on a similar, clear cabin pressure dependency for their $N_2O$,

CO$_2$ and CO measurements (based on an Aerodyne QCLS). They suggested a fast calibration procedure to overcome these dependencies while maintaining a $\geq$ 90 % duty cycle. [...]"

4. **Pg2, L32: the main objective of the paper is missing. What is the final goal of this investigation? Which measurement data and for what purpose are they going to be used? Is the data quality adequate to answer the research questions?**

   We added the missing objective:

   "[...] The instrument is shown to provide airborne observations of high quality at high sampling rate with multi-species sensitivity as required for assessing greenhouse gas fluxes with a regional focus. [...]"

5. **Pg3, Sect 2.1: this section needs some re-work, e.g. statement like "optics compartment contains all optical elements" is redundant, while Laser#1 and #2 without clear definition has no sense. I suggest giving the driving specifications of the lasers (current and temperature) as well as their optical power output. Specify the exact detector type.**

   We have reworked this section to include the above mentioned:

   "[...] The spectrometer is split into an electronics compartment and an optics compartment. The electronics compartment includes an embedded computing system, thermoelectric cooling (TEC) controllers, power supplies, etc.. The optics compartment includes the lasers, the sample cell, the pressure controller, etc..

   Fig. 1 shows a top-view photograph of the optics compartment. A combination of a continuous wave (CW) QCL and ICL measures mixing ratios of $CH_4$, $C_2H_6$, $CO_2$, $CO$, $N_2O$ and $H_2O$ simultaneously by direct absorption spectroscopy. The sample cell is an astigmatic Herriott cell with approximate physical dimensions of 15cm x 15cm x 50cm (WxHxL) made from aluminum. It provides an effective absorption path length of $204\,$m with a net volume of $2.1\,$L. Two laser light sources are tuned to a specific center wavelength by adjusting the temperature using Peltier elements contained in the lasers housing. Excess heat is removed through a liquid cooling/heating circuit (*SOLID STATE COOLING SYSTEMS, New York, USA*). Laser #1 is an Interband cascade laser manufactured by *nanoplus GmbH, Gerbrunn, Germany* with a peak output power of $9.5\,$mW operated at $4.7\,°C$ and modulated between $2988.520\,cm^{-1}$ and $2990.625\,cm^{-1}$ using a linear current ramp of up to $40\,$mA. Laser #2 is a quantum cascade laser manufactured by *ALPES Laser, St-Blaise, Switzerland* with a peak output power of $40\,$mW operated at $1.5\,°C$ modulated between $2227.550\,cm^{-1}$ and $2228.000\,cm^{-1}$ using a linear current ramp of up to $300\,$mA. The lasers are modulated sequentially at a fixed frequency of $1.5\,kHz$. Laser #1 scans over absorption lines of $CH_4$, $C_2H_6$ and $H_2O$, Laser #2 sweeps over $N_2O$, $CO_2$ and $CO$ lines. Each laser is sampled at 450 spectral points. Acquired spectra are co-added to yield a single output spectrum every half of a second. Before reaching the sample cell, the laser beam travels approximately $1.6m$ inside the instrument under ambient conditions. This will be referred to as the open-path of the instrument, which is heavily influenced by variations in cabin pressure, temperature and humidity during airborne operation. After passing through the sample cell, the combined output from both lasers hits a single TEC-cooled detector. A second, identical detector collects radiation from two auxiliary paths. The first auxiliary path contains a small, sealed reference cell filled with $CH_4$ and $N_2O$. This allows for spectral referencing during system startup. The second path introduces an etalon into the beam, allowing for experimental determination of the laser tuning rate, which relates laser supply current and emitted wavelength. [...]"

6. **Pg4, L1: obviously there are many hundreds of reflections within the multi-pass cell, which leads to significant decrease of optical power of the laser beam. What is the reflectivity of the mirrors and how much is the out-coupled ICL intensity?**

   The sample cell has not been modified from its original state. The exact reflectivity of the mirrors employed by Aerodyne Inc. is therefore unknown to us just like we did not measure the out-coupled intensity. From our point of view, this quantity is not relevant for the purpose of this manuscript. It is high enough to provide sufficient signal-to-noise ratio on the acquired spectra.

7. **Pg4, L3: I doubt that the laser devices are directly coupled to the Peltiers. There should be a buffer heat-sink between.**

   The laser devices are supplied by the manufacturer in a TO66 housing with the TECs already built in. This is what is

meant with "[...] Peltier elements directly attached to the laser [...]". We modified the relevant sentence accordingly:
"[...] Both lasers are tuned to a specific center wavelength by adjusting the temperature using Peltier elements contained in the laser housing. [...]"

8. **Pg8, L15: the relative frequency changes seem to be the same, which is also illustrated by Fig8. So what is the real benefit for using five different shift parameters?**

Even very subtle perturbations from the co-linearity of these shift parameters result in large changes in retrieved mixing ratios. Apart from that, the shift parameters provide a means for observing spectral stability, that would not be available otherwise.

"[...] This allows to properly model frequency changes and provides a means for observing spectral stability. Typical shift parameters for ground-based operation are given [...]"

9. **Pg9, L1: how large were the temperature fluctuations within the optical compartment? What was their effect on the spectral retrieval?**

The optical compartment is temperature stabilized by means of a recirculating chiller. As a consequence typical temperature fluctuations inside the optical compartment are on the order of $\sim 0.3\,\mathrm{K}$. We did not observe a correlation between retrieved mixing ratios and these fluctuations. We included this in the section on airborne performance:

"[...] We identified temperature fluctuations within $\sim 0.3$ K, pressure changes of up to $\sim 200$ hPa and relative humidity changes of up to 35 % in the instruments optical compartment during this flight. [...]"

10. **Pg9, Fig 9: specify the averaging time of the spectra.**

The averaging time of these spectra is $\tau = 0.5\,\mathrm{s}$. The values have been included in a revised version of the figure.

11. **Pg9, Fig 9: where is the CH4 line in the C2H6 fit-window? What is causing the strong bias in the residual in the CH4 fit-window?**

The bias in the residual of the CH4 window is the remainder of the third-order polynomial used to model the spectral baseline. The relevant sentence on Pg. 10 has been changed to:

"A single adjacent $CH_4$ line, located at $2989.981\,cm^{-1}$ has been included in order to obtain good $C_2H_6$ data even under these challenging conditions."

12. **Pg10, L7: a short clarification should be added why the authors chose this difficult spectral window? The range around 2224.5 cm-1 would, e.g. contain all the species with significantly less spectral interference. The ambition of getting the CO2 along with CO and N2O introduces severe compromises in the achievable spectral sensitivity and selectivity. Adding the fact that the selected CO2 line is not even the main isotopologue and seems to have large systematic bias, I seriously doubt whether this compromise is worthwhile.**

Thank you very much for hinting towards this spectral region. It seems to be a nice alternative. However, this region does neither include the primary CO2 isotopologue and CO2 absorption is smaller by approx. 30%. We do agree that CO seems to suffer less from spectral interference.

13. **Pg11, L9: indicate the precision and accuracy of the generated water vapor mixing ratios. What about hysteresis effects, i.e. humidifying vs. drying cycle?**

The relevant section might mistakenly give the impression that absolute water vapor concentrations are necessary to compute the water broadening coefficient. It is not. The water foreign broadening coefficient has been adjusted iteratively until reported dry-air mixing-ratios of the species of interest remained constant for the set of water vapor mixing ratios given the fixed concentrations in the calibration gas cylinder employed. We do not state how good our water vapor data actually is in terms of absolute values. Instead we just use the measured water vapor data to correct the species of interest. The cell has been flushed for a longer time with a given water vapor mixing ratio setpoint (on the order of minutes). From this we conclude, that the measurement cell was entirely flushed before every new setpoint. Hyteresis effects, i.e. humidifying vs. drying cycle should therefore have no influence.

14. **Pg12, L4: How well can the results obtained at 1 SLPM transferred to the 23 SLPM operation regim? What about simply using an empirical correction factor on the retrieved mixing ratios instead of introducing the broadening**

**coefficient in the fitting procedure?**

We think that the good agreement between the instruments presented in this work provides evidence enough, that the water broadening coefficients obtained at 1 SLPM can be transferred to the 23 SLPM regime. We further think that the water vapor correction approach used in this study reflects the relevant physical processes best. We did not try to use an empirical correction factor instead of the spectral approach, therefore we can't provide a statement on this.

15. **Pg12, Fig.7: it seems that the plot shows the deviation instead of variance. The Allan deviation plot indicates that the instrument drifts already after 30s even though operated under ground-based conditions. During flight, pressure and temperature variations, as well as mechanical vibrations tend to impair the performance of the instrument at even shorter time-scales. Considering the long-path of the optical cell, I wonder whether the authors did observe any correlated noise behavior when changing gas flow through the cell from 1 to 23 SLPM? As such, it would be useful to see the distribution diagram (or at least to give quantitative estimates of their spread) of the calibration gas measurements during flights.**

Thanks for pointing towards the wrong units in this figure. We did not observe a clearly correlated noise behavior when changing from low to high flow rates. The distribution diagram of the calibration gas measurements has been included in a supplement to the manuscript.

16. **Pg13, L9: what is meant by software based frequency lock mechanism?**

The way the frequency lock mechanism is implemented here is that the laser temperature is regulated to compensate for drifts using the spectral shift as the controller input and the current to the Peltiers as controller output. The controller itself is implemented in software on the data analysis computer.

We included this in the text:

"[...] Software based frequency lock refers to a controller regulating the laser temperature to compensate for drifts using the spectral shift as the controller input and the current to the Peltiers as controller output. The controller itself is implemented in software on the data analysis computer. [...]"

17. **Pg13, Fig8: what is the influence of the sudden frequency shift discontinuities on the retrieved mixing ratios?**

Using the spectral shift handling described within this work, the spectral shift discontinuities observed do not have a conspicuous effect.

18. **Pg17, L4: what would be the required isotopic composition of such a CO2?**

The required isotopic composition of such a CO2 would be 98.447% primary isotopologue and 1.079% secondary isotopologue ( $^{13}C^{16}O_2$ )

"[...] In this study we used working standards of synthetic nature from *Air Liquide*. Usually these are produced with $CO_2$ from natural gas & oil combustion processes. We determined the $CH_4$ and $CO_2$ values of each working standard gas cylinder using a NOAA-anchored (Cert.-Nr. CB11361) Picarro G-1301m. This has the drawback that we do not know the isotopic composition of our working standards as its impact had been considered negligible, e.g. (Chen et al., 2010). We only found out during development of JFIT, that the instrument is using a $^{13}C^{16}O_2$ line to derive ambient $CO_2$. We estimate the required isotopic composition of such a $CO_2$ to be 98.447% primary isotopologue and 1.079% secondary isotopologue or $\delta^{13}C = -19.6\,‰$ which seems reasonable according to B. Coplen et al. (2002). Since we are reporting retrieved mole fractions relative to the WMO scale, only the working standard reproducibility contributes to the total uncertainty of $CH_4$. Uncertainty on $CO_2$ is difficult to assess here because of the unknown isotopic composition in our working standards. [...]"

19. **Pg17, L8: give an estimate of the overall calibration gas consumption for the 18 flights and shortly discuss options for optimization.**

The overall calibration gas consumption for the 18 research flights amounts to $\sim 3.5\,m^3$. A reduction in necessary calibration gas aswell as a significant increase in system response times could be achieved by reducing the physical sample cell volume. We included the overall calibration gas consumption estimate in the revised manuscript:

"[...] calibration strategy from Sect. 2.3. This required a total calibration gas amount of $\sim 3.5\,m^3$ (excluding zero air) for the 18 research flights. [...]"

20. **Pg19, L10: as mentioned earlier, it would be useful in this context to show the distribution diagram of the calibration gas measurements performed at every 10 min interval and representing about 10% of the measurement time.**

The distribution diagram of the calibration gas measurements has been included in a supplement to the manuscript.

21. **Pg19, L17: it is somehow unclear what applies: in the previous section (pg18, L2) the authors claim that they were unable to reproduce the cabin pressure dependence, but in the conclusion is argued that the known cabin-pressure dependencies are effectively minimized by frequent two-point calibration.**

The relevant part of Pg18 was misleading, we therefore rephrased to:

"[...] A severe cabin pressure dependence in excess of $0.3\,ppb\,hPa^{-1}$ in $CH_4$ mixing ratio has been previously reported for airborne TILDAS instrumentation (Pitt et al., 2016). This instrumentation however physically differs from the one reported here. It is not possible to accurately compare the dependencies of one instrument relative to another since many factors/quantities involved are instrument-specific, e.g. the open-path length, the positioning and properties of optical elements, like windows and mirrors, the stiffness and thermal expansion coefficients of employed optical stands, etc.. We were nevertheless able to effectively minimize cabin pressure dependencies during operation of the QCLS instrument aboard the C130 using the calibration strategy from Sect. 2.3. [...]"

22. **Pg19, L24: was the frequency lock mechanism active during flight operation only? Do the frequency-"jumps" correlate with laser heat-sink temperature changes?**

The frequency lock mechanism is always turned on following an initial referencing after instrument startup. We do not see a correlation with laser operating temperature. We did further not directly measure the laser heat-sink temperature nor did we log the recirculating chillers temperature either. This however is a very interesting question with respect to the source of the observed discontinuities that we will try to address soon.

23. **Pg20, L2: Having an uncertainty of 1 ppm and systematic bias of 10 ppm on the CO2 retrieval, projecting towards isotope ratio measurement is quite steep.**

If you want to express that it will be quite a challenge to reach isotope ratio measurements than we fully agree.

**Technical corrections:**

1. **Pg1, L12: "truly" is not a proper attribute for simultaneous. Remove it.**

"truly" has been removed.

2. **Pg2, L15: check reference, because Santoni et al. used QCLS instead of CRDS**

Here, we are referring to established IR spectrometers in general. The wording has been changed accordingly to:

"[...] used for airborne research, e.g. established IR spectrometers [...]"

3. **Pg2, L20: as above, Richter et al. used DFG instead of QCL**

The sentence has been changed to:

"[...] reported on a custom-built difference frequency generation (DFG) absorption spectrometer [...]"

4. **Pg3, L16: here and across the manuscript add space between value and unit. Also the chemical formula should be always printed in Roman (upright) type (see e.g. IUPAC Green Book).**

This has been implemented in a revised version of the manuscript.

5. **Pg4, L3: avoid using laser diode when referring to ICL/QCL devices.**

This comment has been taken into account in the revised version of the manuscript.

6. **Pg8, L4: replace "micro" by "fit" window.**

Occurences of "micro window" have been replaced with "fit window" in a revised version of the manuscript.

7. **Pg21, references: check for typos and completeness, e.g. at L10, L13, L22, etc.**

The reference pages have been checked for typos and completeness in a revised version of this manuscript.

**In response to the Associate Editors comments from February the 14th, 2019.**

*Dear Authors, Thank you for your detailed and authoritative responses to the referees' comments. I have carefully reread your well-written paper that certainly merits publication in AMT. At this stage, there are some very minor and technical issues remaining, which should be addressed before publication. Please find these listed below.*

5 Dear Associate Editor,

Thank you very much for your detailed corrections and the careful look over the manuscript. We greatly appreciate your help with this paper. The technical corrections have been implemented in a revised version of the manuscript.

**Minor corrections:**

(page and line numbers refer to the manuscript version that keeps track of revision changes)

10 1. **Some company names appear in capital letters, others do not. It seems preferrable to keep a common format. I suggest capitalizing only the first letter in company names, which is an easier read, especially for long names.**

We have tried to keep everything in a common format now.

**In the same veins, there is excessive use of the brand name to denote the CRDS analysers, but other instruments such as the modified QCLS/ICLs instrument are not presented by insisting on the original brand each time.**
15 **This poses the risk of inadvertent advertising of one particular instrument as compared to others and hinders easy understanding for people outside the field. In general, it is preferable to use brand/model names only when instruments are presented for the first time and when required for the understanding. Otherwise instruments should be referred to by their operational principle or instrument function (eg "hygrometer" instead of "Sensirion", etc.). Please modify Figures 10, 11, 12, 13 and the text accordingly.**

20 We have modified the text and the figures as suggested.

2. **The typesetting guidelines of Copernicus journals require a space between quantity and unit. This also applies to dimensionless units such as the % sign (ie 78 % instead of 78%, ...).**

We hope to have modified the text accordingly at all occurences of values/units.

3. **p 7, l 17-18 : " ... CH4 (Cert.-Nr. CB11361, WMO X2004A for CH4 (Dlugokencky et al., 2005)). C2H6, CO and**
25 **N2O are compared to NOAA ..." -> CO2 is missing in the list even though it is mentioned later on in chapter 5, p 17. Please complete the list.**

$CO_2$ has been added to the list.

4. **p 16, Fig 8 and discussion of frequency lock : There is a regular spike pattern in Fig 8 with a period of about 30/6 5 min. This supports the hypothesis that shifts are not only caused by mere laser drifts, but are also caused by**
30 **some well-timed mechanism. It is not clear whether this is meant by "high-frequency shifts are evident, including discontinuities". The discussion of the spikes could be added to the discussion on p 16.**

We have included the missing reason for these well-timed spikes. These are due to switching over from calibration to sample gas.

5. **p 22, l 20-21 and p 23 Fig 13 : According to the figures, the QCLS-CRDS and the QCLS-FLASK comparison shows**
35 **a good agreement in relative deviations, except for CH4, where QCLS-CRDS gives a 1-sigma of 1.4 ppb, but the flask comparison seems to give a different number (1-sigma = 3.9 ppb (n=40)). This needs to be commented.**

This could be due to different sampling times of the fast QCLS and CRDS observations compared to the slow flasks. We have added this in the main text.

6. **p 21, l 12+, discussion 13CO2 measurement : "We estimate the required isotopic composition that could explain**
40 **a large bias of up to 17 ppm (see Supplement Section 3) in CO2 to be 98.447 % primary isotopologue and 1.079 % secondary isotopologue or $\delta$13C = -19.6 ‰ which seems reasonable according to B. Coplen et al. (2002)." This**

**statement seems to be in direct conflict with the Supplement, where it is derived that d13C must be -37 ‰ to have an impact of 17.2 ppm. Please correct and be more detailed about the -19.6 ‰ and your calibration procedure that should be crucial in infering a particular isotope composition.**

Thank you for pointing us towards this error. The derivation in the Supplement is correct. Here, we wanted to refer to the 10ppm bias found in this study.

**The discussions in the main text and in the supplement also merit clarification. In particular, the definition of x_retrieved remains dubious, as remains the meaning of "possible influence estimate". It might help to differentiate between two effects of opposite sign : HITRAN conventions (a -5 ppm effect when measuring background air CO2) and the systematic bias that could play a role here (up to about +12 ppm). HITRAN assumes 13C isotope abundances of VPDB (d13C = 0) to derive line intensities, and, for the purpose of demonstration, one assumes that these are correct. When we then measure CO2 as 13CO2 we can use the rule of thumb that a relative change of 13C will translate into a relative change of CO2 even if the concentration of CO2 remains unchanged (less 13C leads to less CO2 measured and higher 13C leads to higher values of CO2). This means that the QCL measurement of air with d13C = -9.7 ‰ leads to CO2 that is roughly 1 % (or 4 ppm at 400 ppm CO2) too low (effect of -4 ppm). A positive offset of +10 ppm can only be obtained if the instrument is calibrated with a gas of d13C = -40 ‰. This seems to be reasonable provided the tank CO2 derives from fossil fuel. Air, which has a d13C that is higher than fossil fuel CO2 by 30 ‰ will yield a 0.03 * 400 ppm = +12 ppm higher CO2 abundance. Note also that line 15 on p 3 of the Supplement wrongly refers to methane.**

We tried to change the wording of x_retrieved and "possible influence estimate". To our knowledge HITRAN does not assume 13C isotope abundances of VDPB, instead the natural terrestrial abundances used in HITRAN which are available on the official website are used. Unfortunately we do not quite understand the derivation given above. We tried to give a detailed derivation in the supplement, that we think is complete as is.

**To which degree has the 13C isotopic composition of CO2 to be known to reach WMO compatibility ?**

The requested info has been appended to the supplement.

**Technical corrections:**

(page and line numbers refer to the manuscript version that keeps track of revision changes)

1. **p 1, title** : use subscripts in chemical formulae
   Changed accordingly.

2. **p 1, abstract** : "... and central US." -> "... and central US." (note you use USA without full stops later on ....)
   Changed accordingly.

3. **p 3, l 5** : "since the pre-industrial era, where ..." -> consider replacing "where" by "and"
   Changed accordingly.

4. **p 3, l 11** : "Aircraft provide ..." -> "Aircrafts provide" (eventually "aircraft provides")
   We refer to the plural of Aircraft here. We think it should be Aircraft not Aircrafts.

5. **p 4, l 25** : consider replacing "... required to operate the instrument on research aircraft." by "... required to operate the instrument on a research aircraft."
   Changed accordingly.

6. **p 4, l 27** : "The spectrometer is split into an electronics compartment and an optics compartment." -> "The spectrometer is split into an electronics and an optics compartment.
   Changed accordingly.

7. **p 4, l 28** : "... , etc.." -> "... , etc."
   Changed accordingly.

8. **p 5, l 3** : "... , etc.." -> "... , etc."
Changed accordingly.

9. **p 6, l 2** : "every half of a second." -> "twice per second."
Changed accordingly.

10. **p 6, l 17** : "... avoiding injecting large vibrations into the ..." -> "... reducing vibrations of the ..."
Changed accordingly.

11. **p 6, l 18** : "This translates to a net flow rate of 25 SLPM". Different entities use different standard conditions (IUPAC, NIST, EPA ...) SLPM is therefore not a well defined unit. It would help to recall "your" standard conditions here (just once in the text). Even better, you could (just once) give the molar flow rate in mole/s here. Note that since 1982 IUPAC
 defines T = 273.15 K and p = 100 kPa as standard conditions.
Changed accordingly.

12. **p 6, l 19** : "when operating with a cell pressure" -> "when operating at a cell pressure"
Changed accordingly.

13. **p 7, l 3** : "Polytetraflouroethylene" -> "polytetraflouroethylene"
 Changed accordingly.

14. **p 7, l 16** : "cross-calibrated using a Picarro CRDS against NOAAA standards" -> "cross-calibrated against NOAAA standards using a CRDS "
Changed accordingly.

15. **p 7, l 24** : "Typical calibration distribution diagrams ..." sounds like a technical term. It could be easier to understand
 using "Histograms of typical calibration measurements are provided ..."
Changed accordingly.

16. **p 8, legend Fig 2** : A letter "H" appears in the drawing, which is not explained in the Fig legend.
Removed clipping of information on this figure

17. **p 10, legend Fig 4** : change all chemical names to small letters, add axis title to x-axis. What is the difference between
 quantities and units used on the y-axis in this and in the previous (Fig 3) figure ? Shouldn't they have same names and spellings ? Since "adu" does not refer to a proper name, it should be spelled using small letters.
Changed accordingly.

18. **p 11, legends and axis labels Fig 5** : font size is too small
Changed accordingly.

19. **p 11, Fig 5, y-axis label** : The plot does not show the optical depth but the transmittance
Changed accordingly.

20. **p 11, Fig 5, graph labels** : replace "microwindow" by "fit window"
Changed accordingly.

21. **p 11, l 6** : "... shift variables are held constant at the mean over its last 10 values." -> "... shift variables are held constant
 at their means over the last ten values."
Changed accordingly.

22. **p 13, l 6** : "Eq. (1)" instead of "Eq. 1"
Changed accordingly.

23. **p 13, Table 1** : Broadening coefficient ($\gamma_{air}$)
 Changed accordingly.

24. **p 15, l 2** : "Power drawn from the aircraft remained under 50 A at all times and settled at approximately 40 A." Replace "power" by "current" or "electrical current".
Changed to "Electric current ..."

25. **p 15, l 3** : "The volumetric flow rate at ... SLPM". You specify the flow of an absolute amount of gas and not a volume flow. I suggest dropping the term "volumetric".
Removed "volumetric".

26. **p 15, l 9** : "will decrease the signals standard deviation" -> "will decrease the standard deviation of most of the signals"
Changed accordingly.

27. **p 15, Fig 7 left panel** : Legend is difficult to read, please increase font size. Plot could be made more readable by excluding data for $\tau > 500$ s (does not seem significant anyway) and restrict y-scale from 1e-4 to 1e1.
We increased font size of the legend.

28. **p 15, Fig 7 right panel** : It appears that the confidence interval given for linear fit parameters ignores the uncertainties of the x-values. If this is so, it is better not to insist on the confidence intervals of the parameters. Give R-squared with number of digits required to see the deviation from 1.
Changed accordingly.

29. **p 16, Fig 8** : Please increase the thickness of lines in legends. Traces are difficult to identify otherwise.
Changed accordingly.

30. **p 20, l 2** : "(Gordon et al., 2017))" -> "(Gordon et al., 2017)"
Changed accordingly.

31. **p 20, l 15** : put "H2O" in upright letters
Changed accordingly.

32. **p 20, l 18** : Please add + or - sign to bias constants in order to avoid ambiguities, assuming that "+" means that the QCLS is higher than the reference.
Changed accordingly.

33. **p 21, l 22** : "(see 2.3)" -> "(see Section 2.3)"
Changed accordingly.

34. **p 21, l 23** : "Eq. (1)" instead of "Eq. 1"
Changed accordingly.

35. **p 21, l 27-28** : "... precision (uncertainty) estimates based on ambient measurements at stable conditions of 0.3 (2) ppb, 0.02 (0.1) ppm and 2.0 (5) ppb ...". This notation is non-standard and quantities in parentheses might therefore be confused with uncertainties. I suggest writing "... precision/uncertainty estimates based on ambient measurements at stable conditions of 0.3/2.0 ppb, 0.02/0.1 ppm and 2.0/5.0 ppb for ...".
Changed accordingly.

36. **p 23, Fig 13** : y-axis label "Norm occurrence" should come with units in 1/ppb or 1/ppm
Changed accordingly.

37. **p 23, Fig 13, central panel** : symmetrize y-axis (from -2.x to +2.x)
Changed accordingly.

38. **p 23, Fig 13 caption** : Please add legend for colour code of flask samples.
Changed accordingly.

39. **p 23, Fig 13 middle panel on top** : Is there a reason for the skew in the distribution ?
Changed accordingly.

40. **p 24, l 2** : "(5 to 10mins interval)" -> "(5 to 10 min interval)"
Changed accordingly.

5   41. **p 24, l 10** : "above the eastern U.S.." -> "above the eastern US."
Changed accordingly.

[revised manuscript text omitted]